

# LandInG 1.0: A toolbox to derive input datasets for terrestrial ecosystem modelling at variable resolutions from heterogeneous sources

Sebastian Ostberg[1], Christoph Müller[1], Jens Heinke[1], and Sibyll Schaphoff[1]

[1]Potsdam Institute for Climate Impact Research (PIK), Member of the Leibniz Association, Potsdam, Germany

**Correspondence:** Sebastian Ostberg (ostberg@pik-potsdam.de)

**Abstract.** We present the Land Input Generator (LandInG) version 1.0, a new toolbox for generating input datasets for terrestrial ecosystem models (TEM) from diverse and partially conflicting data sources. While LandInG 1.0 is applicable to process data for any TEM, it is developed specifically for the open-source dynamic global vegetation, hydrology and crop growth model LPJmL (Lund-Potsdam-Jena with managed Land).

The toolbox documents the sources and processing of data to model inputs and allows for easy changes to the spatial resolution. It is designed to make inconsistencies between different sources of data transparent, so that users can make their own decisions on how to resolve these, should they not be content with the default assumptions made here.

As an example, we use the toolbox to create input datasets at 5 and 30 arc minutes spatial resolution covering land, country, and region masks, soil, river networks, freshwater reservoirs, irrigation water distribution networks, crop-specific annual land use, fertilizer, and manure application. We focus on the toolbox describing the data processing rather than only publishing the datasets as users may want to make different choices for reconciling inconsistencies, aggregation, spatial extent or similar. Also, new data sources or new versions of existing data become available continuously and the toolbox approach allows for incorporating new data to stay up-to-date.

## 1 Introduction

Models describing land surface processes such as terrestrial ecosystem models (TEM) typically require a number of inputs beyond weather variables. While there is a growing trend to publish model source code to strengthen open science, published code often does not include input files, hampering the applicability of published models. Furthermore, TEMs are increasingly run at varying spatial resolutions, often depending on the spatial extent of the application for global, continental or regional-scale simulations, increasing the need for flexibility in input data creation. TEMs like LPJmL (Lund-Potsdam-Jena with managed Land) (Schaphoff et al., 2018b; Von Bloh et al., 2018; Lutz et al., 2019) typically run on a regular spatial grid. Spatially heterogeneous inputs typically need to be provided for that grid, both in terms of spatial resolution and geographic extent, but may be provided originally in the form of national statistics, polygons or other grids. Section 2 describes the different types of input used by LPJmL (Table 1), the source datasets used to create these inputs and the methods used to process the source datasets. The toolbox does not cover weather or climate inputs, as there are ample methods for weather data processing elsewhere (e.g.





**Table 1.** Types of input datasets required by LPJmL

| Input | Description in section |
| --- | --- |
| Climate | Not covered here |
| Grid | Section 2.1 |
| Country and region assignment | Section 2.2 |
| Soil | Section 2.3 |
| Hydrology | Section 2.4 |
| Land use and land management | Section 2.5 |

Lange, 2019). In Section 3 we show results of applying the toolbox to create model inputs at two different spatial resolutions: 5 arc minutes (5') and 30' longitude by latitude. Most of the data processing is conducted with the 'R' open-source software environment for statistical computing and graphics (R Core Team, 2019) and makes extensive use of R extension packages such as 'ncdf4' (Pierce, 2019), 'raster' (Hijmans, 2020), 'rgdal' (Bivand et al., 2019), 'sf' (Pebesma, 2018), and 'lwgeom' (Pebesma, 2019) for processing geospatial data (both in polygon and raster formats), 'foreach' (Microsoft and Weston, 2019), 'doMPI'
(Weston, 2017), 'doParallel' (Microsoft and Weston, 2022), and 'Rmpi' (Yu, 2002) for parallelized computing on multiple CPUs, 'udunits2' (Hiebert, 2016) for unit conversions, and 'stringi' (Gagolewski, 2019) for character string processing.

## 2 Types of input datasets

### 2.1 Land-sea mask and grid

If not taken from an external data source, the definition of the land-sea mask is the primary step and all other input datasets
need to be converted to this land grid. The purpose of the land-sea mask is to tell the model which grid cells to simulate. The land-sea mask should only exclude oceans, but include inland water bodies such as lakes, rivers, and reservoirs, because their inclusion in the simulation is required for lateral processes, such as the routing of water through the river systems, and the computation of evaporation fluxes from water surfaces into the atmosphere.

   For many years, the land-sea mask and spatial resolution of the Climatic Research Unit's (CRU) time-series datasets of high
resolution gridded month-by-month variations in climate (CRU TS, Harris et al., 2020) have determined the default land area and grid size of TEM simulations, e.g. in the Inter-Sectoral Impact Model Intercomparison Project (ISIMIP, Frieler et al., 2017) and also in simulation experiments with LPJmL (Schaphoff et al., 2018a, b). CRU TS data have a spatial resolution of 30' and cover the whole land area except Antarctica (67420 grid cells). However, CRU TS data include some grid cells in the oceans that do not seem to contain any land, while also missing a few grid cells that are covered by inland water bodies.
In order to create gridded land-sea masks at any spatial resolution we derive them from vector polygons instead of existing gridded datasets. There are a number of sources providing global land polygons all of which can be used in principle (e.g. Natural Earth, 2018; OpenStreetMap, 2020; Wessel and Smith, 1996). It should be noted that coastlines vary between these





data sources so the choice of land polygon will affect the derived grid. A smaller polygon covering only a continent or country can be used as well to derive a grid for smaller spatial domains. We use polygons from the Database of Global Administrative

Areas, version 3.6 (GADM, 2018) to derive the land-sea mask to ensure that each grid cell can also be assigned to a country (see section 2.2).

Common rasterization algorithms in geographic information system (GIS) software assign polygon values to a raster cell only if the polygon covers the center of the raster cell which causes misassignments along polygon borders (coastlines). The land-sea mask for LPJmL should not only distinguish land cells from ocean cells but should also provide information on the

fraction covered by land polygons in each cell. To achieve these two goals, we first create a global raster map at the desired target resolution covering all land and ocean cells (e.g. 4320 x 2160 cells at 5' spatial resolution). Each grid cell is assigned a unique ID before converting the grid cells into polygons and conducting a polygon intersection with the GADM dataset of country and state polygons. On the one hand, the intersection operation removes all grid cell polygons that do not intersect with land. On the other hand, each intersection between one or several country polygons and a grid cell can be uniquely assigned

based on the grid ID. Depending on the spatial resolution of the grid cells and the complexity of the country polygons, the intersection operation can take a long time to process which is why the operation is parallelized to run on multiple CPUs. Parallelization is done using the 'foreach' package (Microsoft and Weston, 2019), which supports several parallel backends. The total land area in each grid cell can be calculated by summing up all land polygon areas with the same unique grid ID. All grid cells containing land are included in the grid input file, which provides a list of coordinates of the center points for all grid

cells included in the input dataset. The toolbox allows to set a threshold for a minimum land area to skip grid cells with very small land shares. The threshold defaults to 1000 $\mathrm{m}^2$ and can be changed by the user. Antarctica is also excluded by default. The grid is a static input that does not change over time. In addition to the grid input file the toolbox also creates a list giving the land fraction in each grid cell.

## 2.2    Country and region mask

All grid cells are assigned to a country (admin level 0) in order to allow for the use of country-specific parameters in the model. Seven large countries are further subdivided into states or provinces (admin level 1 for Australia, Brazil, Canada, China, India, Russia, USA). Grid cells are also assigned to administrative units to provide guidance for disaggregating admin-level source data used in this toolbox to create other inputs. We use GADM version 3.6 (GADM, 2018) as the basis of administrative areas. We acknowledge that country borders are disputed in some regions of the world. GADM version 3.6 was chosen because it is

publicly accessible, frequently updated, and covers several administrative levels. However, any other vector-based dataset of administrative areas can be used instead. Some country definitions also change over time, e.g. the former Soviet Union splitting into several independent countries. The country and region input file is static and represents administrative boundaries of circa 2018.

As mentioned in Section 2.1 rasterization algorithms typically select the value of that polygon which covers the center of

the raster cell. This can cause problems along country borders. Our approach assigns to each grid cell the country which has the largest area share within the cell. If a cell contains polygons of a country that is further subdivided into admin level 1, we





first determine the dominant country, then determine the level-1 administrative unit with the largest area share that belongs to the dominant country. This preserves administrative hierarchies. As a special case, GADM includes the Caspian Sea as a separate level-0 administrative unit. We only assign the Caspian Sea to grid cells which are fully covered by it. All cells which

are only partially covered by the Caspian Sea are assigned to the largest admin-0 level that is different from the Caspian Sea. The toolbox provides an option to prescribe a predefined grid such as the one used by the CRU time-series datasets. In this case, any grid cell not covered by any land according to GADM is assigned to a dummy country 'No land'. In addition to the country and region dataset, which provides the dominant administrative unit in each cell, the toolbox also creates a dataset of the number of countries in each grid cell and a dataset that provides grid cell assignments for administrative levels 0, 1, and

2. These two datasets are used to identify grid cells containing country borders and for spatial gap-filling in the land use data processing (see Section 2.5.3 and 2.5.6).

## 2.3 Soil

TEMs such as LPJmL require soil information for all simulated grid cells. The soil characteristics required vary by model. All versions of LPJmL require gridded information about the soil texture class, following the USDA soil texture classification

(Soil Science Division Staff, 2017). Starting with LPJmL version 5, the model also requires a gridded dataset of soil acidity (pH). There are many datasets providing soil information at global or regional scales with various spatial resolutions, such as the WISE derived soil properties on a 30 arc-second grid (Batjes, 2016) and earlier versions at 5' and 30' resolutions (Batjes, 2005, 2012), or the SoilGrids250m dataset (Hengl et al., 2017). The exemplary implementation for LPJmL uses data from the Harmonized World Soil Database (HWSD, FAO/IIASA/ISRIC/ISSCAS/JRC, 2012). HWSD consists of a raster soil map

at 30 arc-second resolution containing only mapping unit IDs for each grid cell and a separate soil attribute database linking attributes to the mapping unit IDs. Each cell in the map has one associated mapping unit, while each mapping unit may be assigned to several cells and may contain one or several soil units. Only top-soil parameters are extracted from HWSD and used for the full soil column here, for application in LPJmL. Sub-soil parameters are also available in HWSD, but are not used here. Two alternative methodologies have been proposed for aggregating soil texture data to a coarser resolution: 1) averaging

the amount of sand, silt, and clay across cells at the source resolution and then deriving the texture class at the target resolution from the averaged shares; 2) determining the texture class in each cell at the source resolution and then selecting in each target cell the texture class with the maximum number of cells at the source resolution (Koirala, 2011). The first methodology was used for previous model versions of LPJmL (see e.g. Schaphoff et al., 2013). However, it may lead to combinations of sand, silt, and clay content at the target resolution that do not exist anywhere in that grid-cell. The problem of unrealistic aggregated

value combinations becomes even more relevant when extracting additional soil characteristics such as soil pH from the source data. Therefore, the toolbox implements an approach based on selecting soil parameters for the target resolution that represent the most prevalent soil at the source resolution. When determining the soil texture class at the target resolution, the toolbox first groups all soil units found in the source cells by soil texture class and mapping unit and determines the texture class with the largest overall area share at the target resolution. For soil units classified as 'Rock outcrops' and 'Glaciers' in the source

dataset, the toolbox provides an option to either treat them like any other soil (the default setting) or to assign them at the target





resolution only if no other soils are present in the cell. Soil units classified as 'Dunes & shift.sands' in the source dataset are re-assigned to the soil texture class 'sand'. Given that not all soil parameters are available for all soil units in HWSD, additional soil parameters besides soil texture class are derived with decreasing priority from 1) the soil unit with the largest area share that belongs to the dominant soil texture class and the dominant mapping unit; 2) the soil unit with the largest area share that belongs to the dominant soil texture class; 3) the soil unit with the largest area share that belongs to the dominant mapping unit. Cells whose texture class is derived from the HWSD special classes 'Rock outcrops', 'Glaciers', and 'Dunes & shift.sands' are assigned a default pH value of 7 if no other information on pH values is available in the cell. If any cells in the target grid have no information in the HWSD source data, they are filled with soil information from surrounding cells. Cells with missing HWSD data usually point to differences in the land-sea mask. This includes cells along coastlines, remote islands, but also large inland water bodies. A search window around the missing cell is expanded in all directions until at least one cell with HWSD source data is encountered. The soil texture and soil pH of the missing cell is then derived as described above using all HWSD source data within the search window. By default, an inverse distance weighting is applied to all source cells within the search window. The power parameter of the weighting function can be changed by the user, including switching off inverse distance weighting completely. We acknowledge that gap-filling missing cells with soil information from surrounding cells introduces uncertainty into the derived dataset at the target resolution, especially for remote islands where the closest HWSD source cell may be thousands of kilometres away. Considering the limited number of affected cells we consider this uncertainty acceptable for global analyses. Users may choose to reduce the maximum search radius (default: 100 degrees) and fill remaining missing cells by other means. The toolbox also provides an option to set cells with missing soil information to a soil code of zero, which will cause LPJmL to skip those cells in simulations.

## 2.4 Hydrology

### 2.4.1 River routing

River routing describes the lateral flow of water between cells (Rost et al., 2008). The river routing input provides for each cell one downstream cell that water is drained to as well as the distance to that downstream cell. This means that any cell in LPJmL may have zero, one or several cells draining into it but may only have no or one cell that it drains to. The LPJmL source code includes a utility to convert a drainage direction map (DDM) into the main river routing input given a grid file. There are a number of global drainage direction datasets, such as DDM30 (Döll and Lehner, 2002) and STN-30 (Vörösmarty et al., 2000) at a spatial resolution of 30', or HydroSHEDS and MERIT Hydro at resolutions down to 3 arc-seconds (Lehner et al., 2008; Yamazaki et al., 2017). The toolbox does not provide functionality to resample an existing DDM to a new resolution so the user needs to provide a dataset at the correct resolution. The HydroSHEDS technical documentation provides some guidance on upscaling a DDM (Lehner, 2013). Eilander et al. (2021) and Wu et al. (2012) also describe upscaling methods and apply them to the MERIT Hydro and HydroSHEDS datasets, respectively. Based on a river routing input file the toolbox derives upstream and downstream cells as well as the upstream catchment area of each cell. Upstream cells are all cells that drain directly or indirectly through intermediate cells to a grid cell. The upstream area of a cell is the sum of the areas of all upstream





cells and the area of the cell itself. The toolbox computes cell areas of each cell based on the latitude coordinate and the grid
resolution assuming the Earth is a perfect sphere and provides a user option to scale cell areas by the land fraction calculated
in Section 2.1. This may provide better estimates of upstream areas in river basins along coastlines under the condition that
land fractions are also accounted for in the TEM simulations. Downstream cells of a cell are all cells that a cell drains to
either directly or through intermediate cells. The end cell in each river basin is either an outlet to the ocean or an inland sink.
Downstream cell lists for each cell are derived by iterating through the river system from each cell to its end cell. Upstream
cell lists and upstream areas are derived by first assigning each cell its own area and then routing cell lists and upstream
areas through the river system like water. This routing starts in cells with zero upstream cells and then travels downstream,
accumulating the cell lists and upstream areas of all cells along the way.

The implementation of irrigation systems in LPJmL allows for constraining the irrigation water by the water that can be
withdrawn from surface water within the same cell and from one neighbouring cell representing conveyance systems and
transportation of water by trucks over limited distances (Rost et al., 2008). Allowing for water transports also mitigates aggre-
gation errors of the routing network. This additional neighbour cell needs to be provided as an input dataset. Since the input
toolbox has no information on actual river discharge or irrigated areas (which both vary during simulation run time) a cell's
upstream area is used as a proxy to distinguish cells with lower or higher discharge within a region. In the original implemen-
tation (Rost et al., 2008), the neighbour cell was defined as the adjacent cell with the largest upstream area. The toolbox allows
to reproduce the original approach using the upstream areas calculated in the previous step. However, varying grid resolutions
affect the maximum transport distance if only allowing adjacent cells to be used as neighbour cells. To account for the effect
of variable grid resolutions, the toolbox also allows to define a search radius (which defaults to 75 km) and select the cell
with the largest upstream area within that search radius. In addition, the toolbox provides an option to exclude upstream and
downstream cells from the search for a neighbour cell. This can be useful to exclude cells from within the same basin, if it does
not make sense to transport water from a neighbour cell connected to the same river. Transport of water over extended dis-
tances implies an associated cost. While not taking actual costs into account, the toolbox supports applying an inverse distance
weighting to the upstream areas in the search radius which favours the selection of a close neighbour in cases where several
potential neighbour cells have similar upstream areas. The power parameter of the weighting function can be changed by the
user, including switching off inverse distance weighting completely. The neighbour search is computationally expensive for
high-resolution grids and large search radii, where each cell may have thousands of potential neighbour cells, which is why the
process is parallelized to run on multiple CPUs.

### 2.4.2   Lakes, rivers

Inland water bodies such as lakes and rivers have surface properties that differ from land as typically simulated in TEMs. The
fraction of each grid cell covered by lakes and rivers is prescribed in LPJmL and needs to be provided as a gridded, static
input. The input dataset is derived from the Global Lakes and Wetlands Database (GLWD, Lehner and Döll, 2004). Level 1
and level 2 of GLWD provide shoreline polygons for lakes, rivers, and reservoirs. Level 3 of GLWD provides a raster map at
30 arc-second resolution of lakes, rivers, reservoirs, and several types of wetlands. The toolbox includes tools to extract grid





**Table 2.** Dam/reservoir parameters for LPJmL input

| Parameter | Description and unit |
|---|---|
| year | first year of operation |
| capacity | maximum storage capacity of the reservoir in $km^3$ |
| area | area of reservoir in cell in $km^2$ |
| hydropower | installed hydropower capacity in MW |
| height | dam height in m |
| purpose | main purpose of dam (hydropower, irrigation, other), secondary purposes |

cell fractions covered by water bodies either from the polygon data (GLWD level 1 and 2) or from the raster data. Processing
of GLWD level 3 raster data is generally faster and allows separate extraction of all types of water bodies included in GLWD,
but it requires that the target resolution is an integer multiple of the source resolution. By default, only lakes and rivers are
used in LPJmL. Processing of GLWD level 1 and 2 polygon data provides more flexibility since it is based on calculating
polygon intersections between the water body polygons and polygons of the grid cells at any target resolution. On the other
hand, levels 1 and 2 of GLWD do not include wetlands, which are part of GLWD level 3. While GLWD source data are used as
an example, both approaches implemented in the toolbox could be modified to apply them to other source datasets providing
similar information. For example, the polygon intersection method could be applied to the HydroLAKES dataset (Messager
et al., 2016), which distinguishes 1.43 million individual polygons of natural lakes and human-made reservoirs, compared to
roughly 247000 lakes and reservoirs included in levels 1 and 2 of GLWD. However, the HydroLAKES dataset excludes areas
covered by rivers, which are included in GLWD and are usually also included in the lake and river fraction input used by
LPJmL.

### 2.4.3 Dams and reservoirs

If running LPJmL with the reservoir module enabled, the model requires two additional inputs: 1) an input describing dams and
reservoirs (Table 2); 2) an input providing the elevation above sea level in each cell. The Global Reservoir and Dam Database
(GRanD, Lehner et al., 2011) is used to create the reservoir input required by LPJmL. GRanD provides geo-referenced dam
locations, polygons depicting reservoir outlines, and a rich attribute table for approximately 7000 large dams ($\geq 0.1$ $km^3$
capacity) globally. Before processing, records, for which no storage capacity is available, are removed from the database.
For dams which have a storage capacity but no reservoir area the toolbox presents an option to fill in an estimated reservoir
area based on a reversal of the area-storage relationship from Lehner et al. (2011, Equation 1 and 2 therein). The toolbox
also provides an option to filter dams based on the 'timeline' attribute, which indicates changes to the status of a dam over
time (e.g. removal or destruction) or dams still under construction. In order to derive an input dataset, dams and reservoirs
need to be assigned to grid cells first. Since the river routing scheme used for modelling is a coarse abstraction of the real-
world river system, simply assigning dams to the grid cell which they would fall into based on their coordinates can lead





to substantial differences between DDM-derived upstream areas (see Section 2.4.1) and catchment areas reported in GRanD. It could essentially assign a dam on a large river to a grid cell adjacent to the river in the modelled river system, or vice versa. Previous approaches therefore sometimes included checking and correcting dam locations by hand (e.g. Biemans et al., 210   2011). The toolbox attempts to automate positioning of dams by optimizing between two terms: 1) the distance between dam coordinates in GRanD and the center coordinate of the assigned grid cell; 2) the deviation between GRanD catchment area and the upstream area of the assigned grid cell. The user must set a maximum search radius (in degrees), which should be selected based on the resolution of the target grid. The user may additionally set a maximum search distance (in metres), which is applied to further constrain the search window. All grid cells $c$ falling into the search window are assigned a combined distance 215   and deviation weight $W_c$:

$$W_c = 1/D^{p_{\mathrm{dis}}} \cdot 1/\Delta A^{p_{\mathrm{dev}} \cdot p_{\mathrm{sign}}} \tag{1}$$

where $D$ is the distance (in metres) between the cell center of cell $c$ and the dam coordinates according to GRanD and $\Delta A$ is the absolute difference between the DDM-derived upstream area of cell $c$ and the catchment area reported in GRanD. The power parameters $p_{\mathrm{dis}}$ and $p_{\mathrm{dev}}$ refer to a distance and area deviation penalty, respectively. In addition, $p_{\mathrm{sign}}$ provides the 220   option to modify the area deviation penalty based on whether the DDM-derived upstream area is larger or smaller than the GRanD-reported catchment area. Using identical values for $p_{\mathrm{dis}}$ and $p_{\mathrm{dev}}$ assigns equal priority to staying close to the original coordinates and getting a good area match, while increasing one parameter over the other can be used to shift relative priorities. The dam is assigned to the cell with the highest combined distance and deviation weight $W_c$ within the search window. The toolbox allows to test several values of $p_{\mathrm{dis}}$, $p_{\mathrm{dev}}$, and $p_{\mathrm{sign}}$ at once and compare their effect on the assigned grid cells. If none 225   of the parameter combinations give satisfactory results the toolbox also allows the user to set manual grid cell assignments for individual dams. Once each dam has been assigned to a grid cell at the target resolution the additional dam and reservoir parameters are assigned (Table 2). The 'Year' column from GRanD is used as first year of operation even though the original data may refer to the year of construction, completion, commissioning, or refurbishment/update (Beames et al., 2019). Columns 'Cap_mcm', 'Area_skm', and 'Dam_hgt_m' from GRanD are used for maximum storage capacity, area of the reservoir, and 230   dam height, respectively. The installed hydropower capacity is a placeholder and is currently set to zero. The first purpose field in the LPJmL input (main purpose) is encoded as '1' for irrigation, '2' for hydroelectricity, or '3' for other purposes, based on the 'Main_use' column from GRanD. The secondary purpose field in the LPJmL input describes whether a dam is also used for irrigation even if it is not its main purpose. It is encoded as '1' if either the 'Main_use' is irrigation or if the 'Use_irri' column from GRanD has a non-missing value. Three additional purpose fields in the LPJmL input are placeholders only and 235   are set to zero.

Due to the technical implementation of dams and reservoirs in LPJmL each grid cell can only contain one dam/reservoir, which cannot change its properties over time. In all cases where several GRanD dams are assigned to the same grid cell their records are merged based on the following rules:

- maximum storage capacities and reservoir areas are summed up across all dams/reservoirs in the cell





– first year of operation of the combined dam is set to the year when at least 50 % of the final total storage capacity is in operation

     – dam height is set to the height of the individual dam with the largest storage capacity

     – main purpose is set to the main purpose with the highest combined storage capacity in the cell

     – use for irrigation (secondary purpose field) is set to '1' if at least one dam in the cell is used for irrigation

Furthermore, the technical implementation in LPJmL always assigns the total reservoir area to the same grid cell that the dam is assigned to. The reservoir area can thus not exceed 100 % of the grid cell area even if the reservoir polygon covers several grid cells. This means that reservoir areas allocated in LPJmL are likely underestimated, especially for smaller grid cell sizes. However, many large-scale reservoirs such as Lake Victoria in Uganda, Lake Baikal in Russia, or Lake Winnipeg and Lake Ontario in Canada are included in the lake and river input described above so that the water surface area is still represented.

If applying the toolbox to create inputs for a different TEM which supports multi-cell reservoirs, the polygon intersection methodology described in Section 2.4.2 could also be modified to extract grid-cell fractions covered by GRanD reservoir polygons. In that case, double accounting of water bodies as lakes in GLWD or HydroLAKES and reservoirs in GRanD would need to be resolved.

     In addition to the input describing dams and reservoirs the reservoir module in LPJmL requires an input of grid cell elevation

above sea level. There are a number of global digital elevation models (DEM), some of which have very high spatial resolutions, such as the Shuttle Radar Topography Mission (SRTM) 1 Arc-Second Global DEM (USGS EROS, 2014), the ALOS World 3D (AW3D30, Takaku et al., 2016), or ASTER Global DEM (NASA/METI/AIST/Japan Spacesystems and U.S./Japan ASTER Science Team, 2019) at 30 m resolution. High-resolution DEMs are commonly distributed in tiles due to the large amount of data, making them resource-intensive to process. TEMs such as LPJmL commonly simulate the land surface at much coarser

resolutions and thus do not require elevation data at such high resolutions. The toolbox uses the ETOPO1 1 Arc-Minute Global Relief Model (NOAA National Geophysical Data Center, 2009; Amante and Eakins, 2009), which was generated from a number of higher-resolution global and regional datasets, to generate an elevation dataset at the target resolution. Processing of ETOPO1 elevation data is conducted using the Generic Mapping Tools Version 6 (GMT6), an open-source collection of command-line tools for manipulating geographic and Cartesian datasets (Wessel et al., 2019). ETOPO1 source data cover

ocean and land areas, which may lead to artefacts when aggregating elevations along coast lines. To avoid such artefacts, the toolbox provides an option to mask out ocean areas from ETOPO1 data before spatial aggregation using the Global Self-consistent, Hierarchical, High-resolution Geography Database (GSHHG, Wessel and Smith, 1996) included with GMT. Spatial aggregation to the target resolution is carried out either by calculating the median (the default setting) or mean elevation across all source data cells in a target grid cell.



## 2.5 Land use and land management

In TEMs such as LPJmL, the surface area in each grid cell containing land (section 2.1) is typically subdivided into tiles covered by natural vegetation, different anthropogenic land uses, and inland water bodies. The spatial extent of water bodies (sections 2.4.2 and 2.4.3) and the spatial extent and composition of anthropogenic land uses is prescribed by input datasets. Often, land use categories represented in TEMs such as LPJmL are croplands and managed grasslands, which can be further disaggregated into areas of specific crops or management types (e.g. irrigation). Other types of anthropogenic land use such as urban areas are not represented in LPJmL and thus not considered here. The remaining part of a cell that is not assigned to water bodies or croplands and managed grasslands is assumed to be covered by natural vegetation. The composition of natural vegetation, i.e. the area share of different plant-functional types (trees, grasses) is determined endogenously by the model and we thus do not provide any input data on the composition of the natural vegetation. The input dataset on anthropogenic land use described here consists of gridded data on (1) separate rainfed and irrigated growing areas of different crops (all of which grow on croplands), (2) the spatial extent of managed grasslands and (3) areas for second-generation biomass plantations. Crop-specific growing areas and the spatial extent of managed grasslands change over time with an annual time step and are based on the combination of multiple source datasets. Figure 1 gives a general overview of the source datasets and the data processing steps. The process outlined in the following sub-sections is for a global dataset at a spatial resolution of 5' covering the time period of 1500–2017. The toolbox does not support creating a land use dataset at a finer spatial resolution than provided by the gridded source datasets. It does support aggregating source datasets to a coarser resolution automatically as long as it is an integer multiple of the source resolution. However, we suggest to follow the process at 5' and aggregate areas to the desired target resolution as the last processing step. The resulting land use dataset does not contain any values for areas of second-generation biomass plantations since the production of second-generation biofuels from grassy or woody biomass is still mostly in the pilot and demonstration stage and thus no global datasets of such growing areas are available. LPJmL simulations can also be run without anthropogenic land use simulating only natural vegetation in which case the land use input dataset described in the following sub-sections is not required.

### 2.5.1 Country-level source data

Country-level source datasets are depicted as an ellipse in figure 1. Time series of crop-specific harvested areas and cropland extent at the country level are taken from FAOSTAT (FAO, 2020b). FAOSTAT data are updated frequently and include annual data for roughly 180 different crops or crop groups and more than 275 countries or country groups from 1961 to close to present-day. FAOSTAT includes a number of countries that ceased or started to exist during its period of coverage (e.g. former USSR, Kazakhstan). Country groups in FAOSTAT can be geographic (e.g. Central America) or socio-economic (e.g. Least Developed Countries). FAOSTAT data do not distinguish between rainfed and irrigated crops. Data on country-level irrigated harvested areas are taken from MIRCA2000 (Portmann et al., 2010) and AQUASTAT (FAO, 2020a). MIRCA2000 distinguishes 26 individual crops or larger crop groups and 402 national or sub-national spatial units, but only provides data for the situation



**Figure 1.** Flowchart of land use (green) and land management (blue) data processing. See section 2.5 for details. Source datasets: GAEZ (IIASA/FAO, 2012), HYDE (Klein Goldewijk et al., 2017), MON (Monfreda et al., 2008), RAM (Ramankutty et al., 2008), AQUASTAT (FAO, 2020a), MIRCA2000 (Portmann et al., 2010), FAOSTAT (FAO, 2020b), LUH2 (Hurtt et al., 2020), MUELLER (Mueller et al., 2012; Mueller, 2012), ZHANG (Zhang et al., 2017).



around the year 2000. As of 2020, AQUASTAT included data for 38 individual crops or crop groups for 167 countries covering the period 1961–2016, but coverage is expanded frequently.

To combine country-level data between the three country-level datasets, countries in FAOSTAT, MIRCA2000 and AQUAS-TAT are matched to the GADM admin level 0 dataset described in section 2.2. Countries included in FAOSTAT that ceased to exist are matched to the closest combination of GADM units, e.g. 'Yugoslav SFR' from FAOSTAT is matched to the combined area of Bosnia and Herzegovina, Croatia, Macedonia, Slovenia, Serbia, Kosovo, and Montenegro from GADM. Sub-national data from MIRCA2000 are aggregated to the corresponding GADM country.

### 2.5.2   Grid-level source data

Grid-level source datasets are depicted as rectangular boxes in figure 1. The following datasets are used to disaggregate country-level data from section 2.5.1 to the targeted spatial resolution. Time series of gridded total, rainfed, and irrigated cropland as well as grazing lands are taken from HYDE version 3.2.1 (Klein Goldewijk et al., 2017). HYDE data cover the whole globe at a spatial resolution of 5'. The temporal resolution is centennial before the year 1700, decadal between 1700 and 2000, and annual for 2000 to 2017. Centennial, decadal, and annual data from HYDE covering the period 1500–2017 are converted from

ASCII grid to NetCDF format, merged across time, and disaggregated to annual values by simple linear interpolation using the Climate Data Operator software (CDO, Schulzweida, 2019). There is a small number of grid cells in HYDE where rainfed and irrigated cropland do not sum up to total cropland and sometimes even exceed total cell area. This is due to a known bug in HYDE 3.2.1 and will hopefully be fixed in the next HYDE release. If inconsistencies are detected an attempt is made to fix them so that rainfed and irrigated cropland in each cell sum up to total cropland and do not exceed total cell area. Gridded

crop-specific harvested areas are taken from Monfreda et al. (2008, referred to as MON). The MON dataset provides static harvested areas for 175 crops at a spatial resolution of 5' representative of the state around the year 2000. The MON dataset does not distinguish between rainfed and irrigated harvested areas. Since the cropland assumptions underlying MON differ from the HYDE cropland used here, the global cropland dataset underlying MON is used as well (Ramankutty et al., 2008, referred to as RAM). The latter provides global cropland extent with no distinction into rainfed and irrigated cropland at a

spatial resolution of 5' representative of the status around the year 2000. Crops in the MON dataset mostly overlap with crop definitions used by FAOSTAT, which is of advantage when combining the two datasets. We note that the Spatial Production Allocation Model (SPAM) also provides gridded, crop-specific harvested areas but distinguishes only 43 crop types (Yu et al., 2020). While not implemented in the current version of the toolbox, the spatial disaggregation methodology described below could be modified to use SPAM or any other data source of choice instead of the MON dataset.

To calculate climatic suitability for multiple cropping, agro-climatic resources are taken from the Global Agro-ecological Zones (GAEZv3.0) database (IIASA/FAO, 2012). The following GAEZ variables are used: frost-free period, reference length of growing period, temperature growing period, thermal climates, Tsum during frost-free period, and Tsum during temperature growing period. The datasets all have a spatial resolution of 5' and are based on mean climatic data for the period 1961-1990.

If replacing any of these source datasets with alternative data sources the toolbox currently does not allow for the spatial

resolution of the derived land use dataset to be finer than the coarsest source dataset.





### 2.5.3 Data processing of country-level land use data

Data processing steps are depicted as rhomboid boxes in figure 1. Grid-cell values of irrigated and total cropland are aggregated to the country level using the country mask generated in section 2.2. This requires that the country mask has been created at the same spatial resolution as the grid-level land use data. Country-level sums of cropland from HYDE should be consistent with

FAOSTAT cropland extent by design (Klein Goldewijk et al., 2017), however, some inconsistencies still exist, especially if running the toolbox at coarser spatial resolutions such as 30' where no grid cells may be assigned to some very small countries, so that these have effectively no cropland to which harvested areas can then be allocated in the next step.

An expanded annual time series dataset of country-level, crop-specific total harvested areas $HA_{\mathrm{crop,tot,country},y}$ is generated by combining FAOSTAT and the MON dataset. Crop names in MON are matched to crop names in FAOSTAT, and MON

harvested areas are aggregated to the country level using the country mask described in Section 2.2. Aggregation of gridded harvested areas can lead to artefacts along country borders since country delineations in GADM, which are used here, do not necessarily match country delineations used to create the MON dataset. Some automatic filtering is done to avoid crop patterns spilling over into neighbouring countries where the crop is not grown: Crop patterns, whose growing areas are located mostly in border cells (i.e. cells with shares of more than one country), are considered artefacts. Also, the MON dataset includes quality

flags describing whether gridded harvested areas are based on national, state or county-level statistics, which are used to derive a minimum share of cropland cells that a crop pattern should occupy in order to not be considered an artefact. Thresholds used in these filters can be fine-tuned by the user but some artefacts may remain. Matching between FAOSTAT harvested areas and MON harvested areas aggregated to the country level results in one of four possible cases:

1. A crop is present in both FAOSTAT and MON although the country sum of MON does not necessarily match the country
value from FAOSTAT. In this case FAOSTAT data for the respective crop and country are added to the expanded time series dataset, and the spatial pattern from MON is marked for usage in the spatial disaggregation according to Eq. 12.

2. A crop is present only in FAOSTAT but is either not a part of the MON dataset or has no harvested areas in the respective country according to MON. In this case, FAOSTAT data for the crop and country are added to the expanded dataset, and the spatial disaggregation follows a simplified approach according to Eq. 13.

3. A crop is present in a country according to the MON dataset but not according to FAOSTAT. This can be one of 17 forage crops and fodder grasses included in the MON dataset but missing completely from FAOSTAT, or a crop that should be included in FAOSTAT yet has no data for the respective country. In this case, the country sum derived from MON is added to the expanded time series dataset for the year 2000, and the spatial pattern from MON is marked for usage in the spatial disaggregation.

4. There are also a few cases where FAOSTAT only has harvested areas for a more general crop group such as 'Nuts, nes' ('nes' denotes crops *not elsewhere specified*) while the MON dataset has harvested areas for individual crops belonging to that group. In this case time series data for the FAOSTAT group crop are disaggregated using the relative share of the





individual crops within the group from the MON dataset. The disaggregated time series is added to the expanded time series dataset and replaces the original FAOSTAT data for the group.

The resultant expanded time series dataset is gap-filled and extrapolated to cover all years of the full time period 1500–2017. Gap-filling and extrapolation are based on the crop-specific cropping intensity $CI_{crop}$ defined as:

$$CI_{crop} = HA_{crop}/CL \qquad (2)$$

where $HA_{crop}$ is the crop-specific harvested area and CL refers to cropland extent. FAOSTAT cropland extent is used for gap-filling and extrapolation wherever available, while HYDE gridded cropland aggregated to the country level is used for years

outside the range covered by FAOSTAT or in case of missing cropland data in FAOSTAT. Data gaps within the country-level time series are filled by first computing $CI_{crop}$ before and after the gap, interpolating $CI_{crop}$ values linearly for the missing years and then multiplying the newly derived values of $CI_{crop,y}$ with cropland extent of the respective year $CL_y$. When extrapolating before (after) the range of years with available harvested area data the first (last) available value of $CI_{crop,y}$ is kept constant. For crops added to the expanded time series from the MON dataset a constant $CI_{crop,2000}$ is used to derive the full time series.

This introduces substantial uncertainty regarding harvested areas of these crops but was considered preferable to dropping them altogether. There are 13 countries in the FAOSTAT database that have cropland but no data at all for harvested areas. These countries are filled automatically using a representative crop mix. For this, $CI_{crop}$ is computed for all crops present in the smallest FAOSTAT country group that includes the country with missing data and then multiplied with country-level cropland extent to derive country-level harvested areas. For example, missing harvested areas in Andorra are filled using the crop mix

of Southern Europe while missing harvested areas in Aruba are filled with the crop mix of the Caribbean. The algorithm keeps track of whether harvested area values in the expanded time series dataset come from the source datasets or have been introduced by gap-filling and extrapolation. This information should be taken into account when assessing the reliability of the derived final land use dataset.

Countries without any crop-specific irrigated harvested areas according to MIRCA2000 but with irrigated cropland according

to HYDE are also filled with a representative mix of irrigated crops similarly to the process described above for FAOSTAT. Irrigated harvested areas from MIRCA2000 are then merged with irrigated crop-specific harvested areas from AQUASTAT. In the current implementation, only data for AQUASTAT crops that can be matched directly to crops in MIRCA2000 are used, all other AQUASTAT crops are discarded. Preference is also given to MIRCA2000 if both datasets provide data for the same crop-country combination in the year 2000. The decision to prioritize MIRCA2000 over AQUASTAT is a choice and

may be changed by other users. Gap-filling and extrapolation of the time series of irrigated harvested areas is conducted as described above for total harvested areas. However, irrigated cropland from HYDE aggregated to the country level is used for all operations since FAOSTAT provides only a time series of total cropland extent, not irrigated cropland extent. Due to the sparsity of source data in MIRCA2000 and AQUASTAT the majority of country-level time series are based on the extrapolation of only one constant $CI_{crop,ir}$ value. Finally, crop groups in MIRCA2000 (e.g. pulses) are disaggregated to individual crops

using the relative share of each individual crop within the group from FAOSTAT.





A number of consistency constraints are enforced on the country-level time series of total harvested areas and irrigated harvested areas derived by the steps above:

1. Total harvested areas of all crops must fit into total cropland in each country, taking into account multiple cropping suitability but not yet taking into account crop-specific spatial patterns.

2. Irrigated harvested areas of each crop cannot exceed total harvested areas of the same crop.

3. Irrigated harvested areas of all crops must fit into irrigated cropland in each country, taking into account multiple cropping suitability.

4. Rainfed harvested areas, which are calculated as the difference between total harvested areas and irrigated harvested areas, must fit into remaining cropland, taking into account multiple cropping suitability.

Constraint 2 ensures plausibility between irrigated and total harvested areas taking into account that (1) irrigated harvested areas are mostly the result of extrapolating based on a constant $\mathrm{CI}_{\mathrm{crop,ir}}$ value using only changes in irrigated cropland while $\mathrm{CI}_{\mathrm{crop,tot}}$ values are more likely to change in FAOSTAT, and (2) rainfed and irrigated cropland may show different trends over time. The toolbox allows the user to choose whether constraint 2 is applied only after gap-filling and extrapolation or before and after gap-filling and extrapolation. Constraints 1, 3, and 4 are in place to ensure that country-level harvested areas can be

disaggregated to the gridded cropland mask. Harvested areas can exceed physical cropland in case of multiple cropping, i.e. harvesting more than once a year from the same piece of land (Waha et al., 2020). We define the upper limit of how many harvests are possible per year as multiple cropping suitability MCS which is computed for each grid cell and separately for rainfed and irrigated crops based on agro-climatic resources from GAEZv3.0 (IIASA/FAO, 2012). The approach is a simplified version of the multiple cropping zones defined in the GAEZ model documentation (Fischer et al., 2012). MCS can take integer

values of one, two, or three denoting single, double, or triple cropping suitability, respectively.

The following sub-sections describe calculations carried out for multi-dimensional variables. The dimensions include space, time, crop, and irrigation system. Table 3 gives an overview of variable names and variable indices used.

Maximum total harvested area threshold per country and year $\mathrm{HAmax}_{\mathrm{all,tot,country},y}$ used for constraint 1 is defined as:

$$\mathrm{HAmax}_{\mathrm{all,tot,country},y} = \sum_{c=1}^{n}(\mathrm{CL}_{\mathrm{rf},c,y} \cdot \mathrm{MCS}_{\mathrm{rf},c} + \mathrm{CL}_{\mathrm{ir},c,y} \cdot \mathrm{MCS}_{\mathrm{ir},c}) \qquad (3)$$

where $\mathrm{CL}_{\mathrm{rf},c,y}$ and $\mathrm{CL}_{\mathrm{ir},c,y}$ are the rainfed (rf) and irrigated (ir) cropland extent in each cell $c$ belonging to the country in year $y$. $\mathrm{MCS}_{\mathrm{rf},c}$ and $\mathrm{MCS}_{\mathrm{ir},c}$ are the rainfed and irrigated multiple cropping suitability in each cell $c$ which are assumed to be constant over time. This estimate of maximum total harvested area is a strong simplification in that it implies that all crops are equally suited for multiple cropping and does not explicitly account for multiple cropping systems that actually exist in the country (Waha et al., 2020). FAOSTAT harvested areas exceeding the threshold can be caused, inter alia, by inconsistencies

between FAOSTAT cropland and HYDE cropland, point to some crops that allow for a higher cropping intensity than GAEZ-derived MCS, or point to errors within FAOSTAT data. For example, FAOSTAT reports four crops with a combined harvested





**Table 3.** Variable names and indices used in calculating crop-specific harvested areas

| Variable | Description |
| --- | --- |
| CI, CIW | cropping intensity, cropping intensity weight |
| CL | cropland (refers to HYDE cropland unless specified otherwise) |
| HA | harvested area |
| HAmax | maximum harvested area threshold |
| MCS | multiple cropping suitability |
| MON | harvested area from Monfreda et al. (2008) dataset |
| RAM | cropland area from Ramankutty et al. (2008) dataset |

| Index | Description |
| --- | --- |
| ir, rf, tot | irrigated, rainfed, total (sum of rainfed and irrigated) |
| rf_on_ir, rf_on_rf | rainfed on irrigated cropland, rainfed on rainfed cropland |
| $c$, country | cell, country |
| $y$ | year |
| crop, all | crop-specific, sum over all crops |

area of almost 13000 ha in Djibouti in 2000 but reports only 1000 ha of cropland, which would imply at least 13 harvests per year and is unreasonable. Finding a case-specific solution for such problems would be beyond the scope of a global toolbox like the one proposed here. In the current implementation whenever the sum of all crop-specific harvested areas in a country

$\mathrm{HA}_{\mathrm{all,tot,country},y}$ exceeds the threshold the harvested areas of all crops are scaled down linearly so that their sum matches $\mathrm{HAmax}_{\mathrm{all,tot,country},y}$. Another possibility that is not currently implemented would be to first scale down harvested area values that have been introduced by gap-filling and extrapolation before modifying values from the source data because the former are considered less reliable than the latter. In addition, each value in the FAOSTAT database is associated with a quality flag that could be used to prioritize keeping some values over others.

Maximum irrigated harvested area threshold per country and year $\mathrm{HAmax}_{\mathrm{all,ir,country},y}$ used for constraint 3 is defined as:

$$\mathrm{HAmax}_{\mathrm{all,ir,country},y} = \sum_{c=1}^{n} (\mathrm{CL}_{\mathrm{ir},c,y} \cdot \mathrm{MCS}_{\mathrm{ir},c}) \tag{4}$$

As for total harvested area, this assumes that all irrigated crops are equally suited for multiple cropping.

    Constraint 4 is applied after constraints 1 to 3 have been applied to total harvested areas and irrigated harvested areas. Rainfed crops can be grown on rainfed cropland but also on irrigated cropland that is not already used for irrigated crops. As

such, the maximum rainfed harvested area threshold per country $\mathrm{HAmax}_{\mathrm{all,rf,country},y}$ is not simply the difference between $\mathrm{HAmax}_{\mathrm{all,tot,country},y}$ and $\mathrm{HAmax}_{\mathrm{all,ir,country},y}$. Since $\mathrm{MCS}_{\mathrm{ir},c}$ and $\mathrm{MCS}_{\mathrm{rf},c}$ are not always the same, $\mathrm{HAmax}_{\mathrm{all,rf,country},y}$ is not simply the difference between $\mathrm{HAmax}_{\mathrm{all,tot,country},y}$ and the sum of all irrigated harvested areas in the country, $\mathrm{HA}_{\mathrm{all,ir,country},y}$, either. Instead it is comprised of two components: (1) The maximum rainfed harvested area on rainfed





cropland per country and year $\mathrm{HAmax}_{\mathrm{all,rf\_on\_rf,country},y}$ is calculated as

$$\mathrm{HAmax}_{\mathrm{all,rf\_on\_rf,country},y} = \sum_{c=1}^{n} (\mathrm{CL}_{\mathrm{rf},c,y} \cdot \mathrm{MCS}_{\mathrm{rf},c}) \tag{5}$$

(2) The maximum rainfed harvested area on irrigated cropland per country and year $\mathrm{HAmax}_{\mathrm{all,rf\_on\_ir,country},y}$ is calculated as the minimum of two terms:

$$\mathrm{HAmax}_{\mathrm{all,rf\_on\_ir,country},y} = \min \left( \begin{array}{c} \sum_{c=1}^{n}(\mathrm{CL}_{\mathrm{ir},c,y} \cdot \mathrm{MCS}_{\mathrm{rf},c}) \\ \mathrm{HAmax}_{\mathrm{all,ir,country},y} - \mathrm{HA}_{\mathrm{all,ir,country},y} \end{array} \right) \tag{6}$$

Finally, the maximum rainfed harvested area per country is the sum of both components:

$$\mathrm{HAmax}_{\mathrm{all,rf,country},y} = \mathrm{HAmax}_{\mathrm{all,rf\_on\_rf,country},y} + \mathrm{HAmax}_{\mathrm{all,rf\_on\_ir,country},y} \tag{7}$$

If the sum of all rainfed harvested areas in a country $\mathrm{HA}_{\mathrm{all,rf,country},y}$ exceeds $\mathrm{HAmax}_{\mathrm{all,rf,country},y}$ this can be solved either by reducing total harvested areas or by increasing the share of irrigated harvested areas while keeping total harvested areas intact. The toolbox prioritizes total harvested areas and therefore expands irrigated harvested areas. However, as shown in equation 6 any increase of $\mathrm{HA}_{\mathrm{all,ir,country},y}$ further reduces $\mathrm{HAmax}_{\mathrm{all,rf\_on\_ir,country},y}$. Therefore, more rainfed harvested areas than the difference between $\mathrm{HA}_{\mathrm{all,rf,country},y}$ and $\mathrm{HAmax}_{\mathrm{all,rf,country},y}$ need to be converted to irrigated harvested areas. A new target value for the sum of rainfed harvested areas of all crops $\mathrm{HA}^{*}_{\mathrm{all,rf,country},y}$ is calculated as:

$$\mathrm{HA}^{*}_{\mathrm{all,rf,country},y} = \mathrm{HAmax}_{\mathrm{all,rf\_on\_rf,country},y}$$
$$+ \sum_{c=1}^{n}(\mathrm{CL}_{\mathrm{ir},c,y} \cdot \mathrm{MCS}_{\mathrm{rf},c}) \cdot \frac{\mathrm{HAmax}_{\mathrm{all,ir,country},y} - \mathrm{HA}_{\mathrm{all,tot,country},y} + \mathrm{HAmax}_{\mathrm{all,rf\_on\_rf,country},y}}{\mathrm{HAmax}_{\mathrm{all,ir,country},y} - \sum_{c=1}^{n}(\mathrm{CL}_{\mathrm{ir},c,y} \cdot \mathrm{MCS}_{\mathrm{rf},c})} \tag{8}$$

A new target value for the sum of irrigated harvested areas of all crops $\mathrm{HA}^{*}_{\mathrm{ir,country},y}$ is calculated as:

$$\mathrm{HA}^{*}_{\mathrm{all,ir,country},y} = \mathrm{HA}_{\mathrm{all,tot,country},y} - \mathrm{HA}^{*}_{\mathrm{all,rf,country},y} \tag{9}$$

When disaggregating $\mathrm{HA}^{*}_{\mathrm{all,ir,country},y}$ to individual crops preference is given to crops that already have irrigated areas while also respecting consistency constraint 2 from above.

### 2.5.4 Spatial disaggregation of country-level land use data

Weighting maps for the spatial disaggregation of country-level crop-specific harvested areas to gridded cropland are derived from the MON and RAM datasets. These weighting maps describe regional differences in the relative shares of cropland occupied by each crop. Crop-specific harvested areas in each cell $c$ of the original MON dataset are based on census data at either national, state/province, or county/district level under the basic assumption that crop-specific cropping intensity $\mathrm{CI}_{\mathrm{crop},c}$ is constant in all cells belonging to the same administrative area (county/district, state/province, or country) (Monfreda et al., 2008). At first, we derive maps of cropping intensity weights $\mathrm{CIW}_{\mathrm{crop},c}$ from MON and RAM as:

$$\mathrm{CIW}_{\mathrm{crop},c} = \frac{\mathrm{MON}_{\mathrm{crop},c}}{\mathrm{RAM}_{c}} \tag{10}$$





where $\mathrm{MON}_{\mathrm{crop},c}$ is crop-specific harvested area from MON and $\mathrm{RAM}_c$ is cropland extent from RAM, both for the year 2000. Due to inconsistencies between cropland extent in the RAM dataset and HYDE cropland extent used here there are more than 300000 5' grid cells which have cropland according to HYDE but no harvested areas of any crop according to the MON dataset. Because neither the original census data nor the administrative area delineations used to construct the MON dataset are available, an attempt is made to emulate the process to gap-fill missing values in the $\mathrm{CIW}_{\mathrm{crop},c}$ maps. All cells are assigned to GADM level 0, 1, and 2 units using the process from section 2.2. For each GADM level 2 unit (county) that has cells with missing harvested areas a fill value is calculated as

$$\mathrm{CIW}_{\mathrm{crop,fill}} = \sum_{c=1}^{n}(\mathrm{CIW}_{\mathrm{crop},c} \cdot \mathrm{RAM}_c) / \sum_{c=1}^{n} \mathrm{RAM}_c \tag{11}$$

using all cells $c$ within the GADM level 2 unit. If $\mathrm{RAM}_c$ equals zero for all cells within the GADM level 2 unit, all cells $c$ within the GADM level 1 unit are used. If the RAM dataset does not have any cropland within the GADM level 1 unit either, all cells $c$ belonging to the country are used for gap-filling. Information on how many administrative units have been gap-filled along with the source of the fill data is provided to the user.

Gap-filled maps of $\mathrm{CIW}_{\mathrm{crop},c}$ are then used together with grid-level time series of HYDE rainfed and irrigated cropland to disaggregate country-level time series of crop-specific rainfed and irrigated harvested areas. For each country grid-level harvested areas are derived as:

$$\mathrm{HA}_{\mathrm{crop,sys},c,y} = \frac{\mathrm{CIW}_{\mathrm{crop},c} \cdot \mathrm{CL}_{\mathrm{sys},c,y}}{\sum_{c=1}^{n}(\mathrm{CIW}_{\mathrm{crop},c} \cdot \mathrm{CL}_{\mathrm{sys},c,y})} \cdot \mathrm{HA}_{\mathrm{crop,sys,country},y} \tag{12}$$

If $\mathrm{CIW}_{\mathrm{crop},c}$ equals zero for all cropland cells $c$ a simplified version is used:

$$\mathrm{HA}_{\mathrm{crop,sys},c,y} = \frac{\mathrm{CL}_{\mathrm{sys},c,y}}{\sum_{c=1}^{n} \mathrm{CL}_{\mathrm{sys},c,y}} \cdot \mathrm{HA}_{\mathrm{crop,sys,country},y} \tag{13}$$

In both equations, sys is a placeholder for system and may be either ir or rf. If there is no rainfed cropland in the country total cropland is used instead to calculate $\mathrm{HA}_{\mathrm{crop,rf},c,y}$, taking into account that rainfed crops are allowed to grow on irrigated cropland.

Once all rainfed and irrigated crops have been disaggregated to the grid the algorithm checks whether multiple cropping limits are respected in all cells:

$$\mathrm{HA}_{\mathrm{all,rf},c,y} / \mathrm{CL}_{\mathrm{rf},c,y} \leq \mathrm{MCS}_{\mathrm{rf},c} \text{ and } \mathrm{HA}_{\mathrm{all,ir},c,y} / \mathrm{CL}_{\mathrm{ir},c,y} \leq \mathrm{MCS}_{\mathrm{ir},c} \tag{14}$$

Any country where at least one cell violates at least one multiple cropping limit is processed again by a spatial redistribution algorithm. This happens quite frequently, especially when harvested areas are close to $\mathrm{HAmax}_{\mathrm{all,ir,country},y}$ or $\mathrm{HAmax}_{\mathrm{all,rf,country},y}$. While the first spatial disaggregation is performed on a crop-by-crop basis spatial redistribution requires that grid-level harvested areas for all crops are loaded at once which leads to substantial memory requirements for this process. The current implementation in the toolbox provides a compromise between overall random access memory (RAM) requirements and limiting the number of file input/output operations which are slow.





At the beginning of spatial redistribution constraints for irrigated harvested areas are enforced in each cell $c$:

$$\text{HAmax}_{\text{all,ir},c,y} = \begin{cases} \text{CL}_{\text{ir},c} \cdot (\text{MCS}_{\text{ir},c} - \text{MCS}_{\text{rf},c}) & \text{if } \text{HA}_{\text{all,rf,country},y} = \sum_{c=1}^{n} \left( (\text{CL}_{\text{rf},c} + \text{CL}_{\text{ir},c}) \cdot \text{MCS}_{\text{rf},c} \right) \\ \text{CL}_{\text{ir},c} \cdot \text{MCS}_{\text{ir},c} & \text{otherwise} \end{cases} \tag{15}$$

where $\text{HAmax}_{\text{all,ir},c,y}$ is the maximum sum of crop-specific irrigated harvested areas in each cell $c$ and $\text{HA}_{\text{all,rf,country},y}$ is the sum of all rainfed harvested areas in the country. This constraint gives preference to irrigated harvested areas on irrigated cropland by allowing them to potentially use all irrigated cropland in all cells unless the country-level sum of all rainfed harvested areas requires usage of all rainfed and irrigated cropland to allocate rainfed harvested areas. Irrigated harvested areas of all crops $\text{HA}_{\text{crop,ir},c,y}$ are scaled down in cells where the threshold is exceeded:

$$\text{HA}^*_{\text{crop,ir},c,y} = \text{HA}_{\text{crop,ir},c,y} / \max\left(1, \text{HA}_{\text{all,ir},c,y} / \text{HAmax}_{\text{all,ir},c,y}\right) \tag{16}$$

For rainfed harvested areas the threshold takes into account possible changes in irrigated harvested areas due to the application of equation 16:

$$\text{HAmax}_{\text{all,rf},c,y} = \text{CL}_{\text{rf},c} \cdot \text{MCS}_{\text{rf},c} + \min\begin{pmatrix} \text{CL}_{\text{ir},c} \cdot \text{MCS}_{\text{rf},c} \\ \text{CL}_{\text{ir},c} \cdot \text{MCS}_{\text{ir},c} - \text{HA}^*_{\text{all,ir},c,y} \end{pmatrix} \tag{17}$$

Rainfed harvested areas of all crops are scaled down in cells where the threshold is exceeded:

$$\text{HA}^*_{\text{crop,rf},c,y} = \text{HA}_{\text{crop,rf},c,y} / \max\left(1, \text{HA}_{\text{all,crop,rf},c,y} / \text{HAmax}_{\text{all,rf},c,y}\right) \tag{18}$$

Spatial redistribution is an iterative process that tries to preserve spatial base patterns provided by the MON-derived weighting maps $\text{CIW}_{\text{crop},c}$ while also ensuring that constraints from equations 15 and 17 are met and the sum over all cells of crop-specific harvested areas still matches the country-level time series. For iteration $i = 1$, $\text{HA}_{\text{crop,sys},c,y,i}$ equals $\text{HA}^*_{\text{crop,ir},c,y}$ from Equation 16 for $\text{sys} = \text{ir}$ and $\text{HA}^*_{\text{crop,rf},c,y}$ from Equation 18 for $\text{sys} = \text{rf}$. In each iteration $i$, irrigated harvested areas are processed before rainfed harvested areas.

In a first step, crop-specific harvested areas are logit-transformed:

$$\text{HAlogit}_{\text{crop,sys},c,y,i} = \ln\left(\frac{\text{HA}_{\text{crop,sys},c,y,i} / \text{HAmax}_{\text{all,sys},c,y,i}}{1 - \text{HA}_{\text{crop,sys},c,y,i} / \text{HAmax}_{\text{all,sys},c,y,i}}\right) \tag{19}$$

This transformation assigns values between $\text{HAlogit}_{\text{crop,sys},c,y,i} = -\infty$ for $\text{HA}_{\text{crop,sys},c,y,i} = 0$ and $\text{HAlogit}_{\text{crop,sys},c,y,i} = \infty$ for $\text{HA}_{\text{crop,sys},c,y,i} = \text{HAmax}_{\text{all,sys},c,y,i}$. In a second step, an increment $\text{incr}_{\text{crop,sys},c,i}$ is added to $\text{HAlogit}_{\text{crop,sys},c,y,i}$ and values are transformed back to derive an increment in harvested areas:

$$\text{HAincr}_{\text{crop,sys},c,y,i} = \frac{\text{HAmax}_{\text{all,sys},c,y,i}}{1 + \exp(-(\text{HAlogit}_{\text{crop,sys},c,y,i} + \text{incr}_{\text{crop,sys},c,i}))} - \text{HA}_{\text{crop,sys},c,y,i}, \text{ where} \tag{20}$$

$$\text{incr}_{\text{crop,sys},c,i} = \begin{cases} 0 & \text{if } \text{HA}_{\text{all,sys},c,y,i} = \text{HAmax}_{\text{all,sys},c,y,i} \\ 1 - \frac{\sum_{c=1}^{n} \text{HA}_{\text{crop,sys},c,y,i}}{\text{HA}_{\text{crop,sys,country},y}} & \text{otherwise} \end{cases} \tag{21}$$





A value of $\mathrm{incr} = 0$ also results in a $\mathrm{HAincr} = 0$. The former is set to zero in all cells where harvested areas of all crops already equal the maximum threshold $\mathrm{HAmax}_{\mathrm{all,sys},c,y,i}$, i.e. there is no space left for expansion. Higher values of $\mathrm{incr}$ reduce the number of iterations needed, but also increase the risk of overshooting the country-level sum $\mathrm{HA}_{\mathrm{crop,sys,country},y}$. In our implementation, $\mathrm{incr}_{\mathrm{crop,sys},c,i}$ approaches zero as the assigned gridded harvested areas approach the country-level sum. Yet,

it is possible that Equation 20 leads to an overshoot which is why $\mathrm{HAincr}_{\mathrm{crop,sys},c,y,i}$ is corrected if needed:

$$\mathrm{HAincr}_{\mathrm{crop,sys},c,y,i} = \mathrm{HAincr}_{\mathrm{crop,sys},c,y,i} \cdot \min\left(1, \frac{\mathrm{HA}_{\mathrm{crop,sys,country},y} - \sum_{c=1}^{n} \mathrm{HA}_{\mathrm{crop,sys},c,y,i}}{\mathrm{HAincr}_{\mathrm{crop,sys},c,y,i}}\right) \tag{22}$$

The corrected harvested area increment is added to currently assigned harvested areas:

$$\mathrm{HA}_{\mathrm{crop,sys},c,y,i+1} = \mathrm{HA}_{\mathrm{crop,sys},c,y,i} + \mathrm{HAincr}_{\mathrm{crop,sys},c,y,i} \tag{23}$$

Even though Equation 20 prevents individual crops from exceeding $\mathrm{HAmax}_{\mathrm{all,sys},c,y,i}$ the simultaneous expansion of several

crops within the same cell may cause their sum to exceed the threshold. This is why the correction described in Equations 16 and 18 is applied to $\mathrm{HA}_{\mathrm{crop,sys},c,y,i+1}$ before proceeding to the next iteration.

The iterative process described above is repeated for irrigated and rainfed crops until the sum of grid-level harvested areas per crop matches the country-level values or until a maximum number of 1000 iterations. The limit of 1000 iterations is set to limit overall run time of the process and may be changed by the user. The number of required iterations can differ for each

crop in a country. The spatial redistribution algorithm includes a number of additional functionalities to enable successful redistribution in special cases:

- Crops that have not been successfully redistributed after 100 iterations are allowed to expand beyond the base pattern given by $\mathrm{CIW}_{\mathrm{crop},c}$. For this, $\mathrm{HA}_{\mathrm{crop,sys},c,y,i}$ in Equation 19 is set to a tiny area in cells where it is zero and where there is still space left for expansion of harvested areas.

- If redistribution of all irrigated crops finishes before redistribution of rainfed crops the algorithm checks whether $\sum_{c=1}^{n} \mathrm{HAmax}_{\mathrm{all,rf},c,y,i} \geq \mathrm{HA}_{\mathrm{all,rf,country},y}$, i.e. whether the allocated patterns of irrigated harvested areas leave enough space to distribute rainfed harvested areas. If not, $\mathrm{HAmax}_{\mathrm{all,rf},c,y}$ is expanded in a separate iterative process similar to the one in Equations 19 and 20. The expansion ensures that $\sum_{c=1}^{n} \mathrm{HAmax}_{\mathrm{all,rf},c,y} \geq \mathrm{HA}_{\mathrm{all,rf,country},y}$ and $\sum_{c=1}^{n} \mathrm{HAmax}_{\mathrm{all,ir},c,y} \geq \mathrm{HA}_{\mathrm{all,ir,country},y}$. Further spatial redistribution of irrigated harvested areas is necessary after

forcing the expansion of $\mathrm{HAmax}_{\mathrm{all,rf},c,y}$.

If any crops have not finished spatial redistribution after 1000 iterations the remaining missing harvested areas are distributed equally to any remaining space within the country in a simple additive approach to ensure that the sum of grid-level harvested areas of each crop matches the country-level values. Besides creating crop-specific spatial patterns the redistribution algorithm also provides extensive diagnostic output regarding the amount of redistributed harvested areas, harvested areas allocated out

of pattern and the number of required iterations for each country and crop.





### 2.5.5 Conversion of crop-specific harvested areas to growing areas

Crop-specific harvested areas for 178 rainfed and irrigated crops created by the spatial disaggregation are aggregated to so-called crop-functional types (CFTs) that are represented in LPJmL by representative crops of these groups. The list of CFTs currently contains 12 individual crops or crop groups (Schaphoff et al., 2018b), however the toolbox allows for easy modifi-
cations of the CFT definitions. All crops that do not belong to one of the 12 CFTs are aggregated to an 'others' category. As part of the CFT aggregation, the toolbox also allows for a spatial aggregation of harvested areas to a coarser spatial resolution. As mentioned before, we suggest to perform data processing outlined in section 2.5.3 and 2.5.4 at the highest possible spatial resolution and only aggregate to a coarser resolution in this final step.

LPJmL currently does not support modelling of multiple cropping systems while the harvested area dataset includes multiple
cropping. Technically, the sum of all land use categories in the LPJmL input dataset (crop-specific growing areas, managed grasslands and second-generation biomass plantations) and any assigned water bodies must not exceed the total grid cell area. In order not to compromise the space available for natural vegetation, which would have implications for simulating the global biogeochemical cycles, it is also desirable that crop-specific growing areas do not exceed the spatial extent of cropland. The toolbox implements a very simple approach to solve this problem and create a land use dataset that can be used with LPJmL.
However, depending on the specific analysis planned by the user, other approaches may be more suitable. In grid cells where the sum of crop-specific rainfed harvested areas exceeds rainfed cropland all rainfed crops are scaled down linearly so that the sum of their growing areas fits into rainfed cropland assuming a multiple cropping intensity of 1. In some rare cases where rainfed harvested areas have been allocated entirely to irrigated cropland in the spatial disaggregation these rainfed crops are lost. Likewise, in grid cells where the sum of crop-specific irrigated harvested areas exceeds irrigated cropland all irrigated
crops are scaled down linearly so that the sum of their growing areas fits into irrigated cropland assuming a multiple cropping intensity of 1. In grid cells where the sum of crop-specific irrigated/rainfed growing areas is smaller than irrigated/rainfed cropland the remaining cropland is declared as irrigated/rainfed fallow land. This is only a rough estimate of fallow land because crops could be grown in multiple cropping systems, leaving even more cropland fallow.

Finally, the 'managed grasslands' category of the LPJmL input dataset is created from the HYDE land use category 'grazing'.
The LPJmL input dataset features a distinction between rainfed and irrigated managed grasslands which in theory allows for the simulation of intensively managed, irrigated pastures. The HYDE dataset does not include information about irrigated pastures. All grazing areas from HYDE are assigned to the rainfed managed grasslands category in the LPJmL input.

Different irrigation systems, as delineated by (Jägermeyr et al., 2015) for LPJmL, are not distinguished here. This disaggregation of irrigation systems is not a requirement to run LPJmL but could be added to this toolbox at a later stage.

### 2.5.6 Nitrogen fertilizer

Starting with LPJmL version 5, the model includes a representation of the nitrogen cycle (Von Bloh et al., 2018). Earlier model versions do not feature a nitrogen cycle and it can also be switched off in LPJmL version 5. If running with the nitrogen cycle and with anthropogenic land use, the model requires nitrogen fertilizer and/or manure as additional input datasets. The





following describes the generation of time series of crop-specific (chemical) nitrogen fertilizer application rates and application
rates of nitrogen included in manure, which are not differentiated by crop. The dataset combines spatial patterns of crop-specific
fertilizer application rates for more than 140 crops for the year 2000 (Mueller et al., 2012; Mueller, 2012) with historical
fertilizer application trends for five aggregated crop types from the Land-Use Harmonization Project (LUH2, Hurtt et al.,
2020). Spatial patterns and historical trends of manure application rates are based on Zhang et al. (2017). All three source
datasets are gridded, although LUH2 data show country data on a grid. Similar to the land use dataset, we suggest carrying
out data processing at the source resolution and aggregating to the desired target resolution as the last step. Both the fertilizer
patterns and the manure data have a spatial resolution of 5' whereas the LUH2 dataset is at 15'.

Both fertilizer datasets show fertilizer application rates only in grid cells where the respective dataset assumed the crop to be
grown. Since crop-growing patterns underlying these datasets differ from each other and from the land use dataset created by
this toolbox, patterns are first gap-filled to provide values in all grid cells. The spatial fertilizer patterns are based on national or
sub-national data. Gap-filling is done similarly to filling the MON dataset described in section 2.5.4. Missing cells are assigned
to GADM level 2, 1, and 0 units and representative values for the unit are derived using the median of non-missing values,
starting at the smallest administrative level and advancing to larger units if no value has been found. Since administrative
units used here do not match units used in the creation of the fertilizer dataset, the user may provide a minimum threshold
of cells that must be present in an administrative unit to use its value (default value: 5). Additionally, international border
cells — i.e. cells containing more than one country (see section 2.2) — are not used for gap-filling to avoid values spilling
over into neighbouring countries. The LUH2 dataset is gap-filled using the country mask included in that dataset. As such,
the administrative unit delineations match the data. The LUH2 dataset uses a value of zero in cells where a crop is not grown.
In order to distinguish between missing cells and cells with an actual application rate of 0, we first filter out all values in
cells where the respective crop growing area is zero according to the LUH2 dataset. Since the LUH2 fertilizer data represent
country values, the toolbox does not fill up the spatial pattern but collects country values in tabular form using the median of
all non-missing cell values in each country. Fertilizer application rates are missing completely for a few countries in both the
spatial patterns and the LUH2 dataset. These are filled using representative rates derived from larger geographic regions and
taking into account development status. Countries are assigned to geographic regions as defined by the Statistics Division of
the United Nations Secretariat (UNSD, 2022). Representative rates are preferably derived as the median of other countries in
the same UNSD Intermediate Region with the same development status, but may also be taken from the UNSD Sub-region or
Region or from countries with a different development status if no value can be found using the stricter search criteria. Datasets
are always filled with values from the same dataset. When combining gridded fertilizer patterns for the year 2000 with country-
scale fertilizer trends from the LUH2 dataset, care has to be taken because both GADM (used as administrative units for the
spatial fertilizer patterns) and LUH2 use a number of non-standard countries, such as the Caspian Sea and The Spratly Islands
in GADM or the former USSR and former Yugoslav Republic used in LUH2. The toolbox includes country replacement rules
for the default source datasets mentioned in this section and allows the user to add or modify country replacement rules easily
in case they use different source datasets. Annual country-scale fertilizer application rates for the five aggregate crop types
included in the LUH2 dataset are converted into relative scaling fractions normalized to the year-2000 rates in that dataset. In





order to derive gridded crop-specific time series, all crops from the Mueller et al. (2012) dataset are assigned a corresponding LUH2 aggregate crop type. This is a simplification because it assumes that, for example, all 62 C3 annual crops included in the Mueller et al. (2012) dataset follow the same historical trend. Annual gridded values for all cells in a country $c$ and year $y$ are then derived by multiplying their year-2000 value with the country-scale LUH2-derived scaling factor of the corresponding group crop, country and year. In a final step, crop-specific gridded time series of fertilizer application rates are aggregated to the spatial target resolution and the CFTs modelled by LPJmL as a weighted mean, using crop-specific harvested areas from the land use dataset detailed in the previous sections as weighting factors for crops and managed grassland areas as weighting factors for fertilizer applied to pastures. Since applying the historical trend of a whole crop-group to individual crops can lead to unreasonably high fertilizer application rates in a few cases, the toolbox allows the user to define an optional maximum application rate (default: 500 $\mathrm{kg/ha}$) and cuts off any values exceeding that threshold.

Manure application rates from Zhang et al. (2017) are expressed as kilogramme per square kilometre cell area and need to be rescaled to the cropland area in each cell. This is done using total cropland from HYDE, which is also used to derive the land use dataset. This rescaling can lead to some very high application rates in case of inconsistencies between the two datasets. The toolbox allows the user to define an optional maximum application rate and cuts off any values exceeding that threshold. If the target resolution is coarser than the source resolution, gridded manure application rates are aggregated to the target resolution by weighted mean using rainfed or irrigated cropland as a weighting factor, respectively. By default, all crops are assigned the same manure application rate.

## 3 Sample application

In this section the toolbox is applied to generate global input datasets at two spatial resolutions: 5' and 30'. We prescribe the land-sea mask of the CRU dataset (Harris et al., 2020) for the dataset at 30' resolution. The resulting datasets are compared among each other as well as to existing comparable datasets. The structure of this section generally follows the structure of section 2.

### 3.1 Land-sea mask, grid, country and region mask

Polygon shapes from GADM version 3.6 (GADM, 2018) are used to derive the land-sea mask. We exclude Antarctica from the land-sea mask at both spatial resolutions. The intersection of GADM polygon shapes with grid cells at 5' spatial resolution returns 2298844 cells with land. Applying a minimum land area of 1000 $\mathrm{m}^2$ per cell, 618 grid cells with a total area of 123120 $\mathrm{m}^2$ are removed, leaving a grid with 2298226 cells. The grid covers a total grid area of 139 million square kilometres. Taking into account the land fraction in each cell, the grid covers a total land area of $135 \cdot 10^6$ $\mathrm{km}^2$ and thus includes roughly 3 % ocean area (Table 4).

The intersection of GADM polygon shapes with grid cells at 30' spatial resolution returns 70391 cells with land, of which 68 cells have less than 1000 $\mathrm{m}^2$ of land area. However, the CRU land-sea mask only consists of 67420 grid cells. 3112 cells with land according to GADM are not included in CRU land-sea mask. On the other hand, the CRU land-sea mask includes





141 cells which do not contain any land according to GADM. The differences between the two land-sea masks are located mostly along the coastlines of all continents and many islands. The GADM-based land-sea mask also includes the Caspian Sea and a few large inland lakes that are excluded from the CRU land-sea mask. The CRU land-sea mask includes a number of cell clusters in the Atlantic and Pacific Ocean that appear to be far off any known landmass. In total, the CRU grid covers a global

grid area of $146.4 \cdot 10^6$ km$^2$, of which $134.5 \cdot 10^6$ km$^2$ are located on land according to GADM (Table 4). This means that the CRU-based grid may include roughly 8 % ocean area. Not accounting for ocean areas included in the grid cells may create a bias when aggregating grid-based terrestrial biogeochemical fluxes of carbon, water, and nitrogen to the global scale.

GADM version 3.6 includes 256 unique level-0 administrative units (mostly countries). We add a dummy unit 'No land' which we assign to cells in the 30' grid that do not contain any land according to GADM. Since we exclude Antarctica from

processing it is missing in the country masks at 5' and 30' spatial resolution. Vatican City (ISO code 'VAT') is too small to be assigned to even a single grid cell at both spatial resolutions, while another 18 GADM level-0 units are too small to be assigned to a single grid cell at 30' spatial resolution. While automatic country assignment results in 141 cells assigned to 'No land' in the country mask at 30' spatial resolution the majority of these cells are actually located directly adjacent to grid cells with a valid country code and could be re-assigned manually by the user.

GADM version 3.6 includes 252 unique level-1 administrative units (provinces, states) in the seven large countries for which LPJmL distinguishes sub-national units. 252 and 242 of these level-1 units are large enough to be assigned to at least one grid cell at 5' and 30' spatial resolution, respectively.

When using national or sub-national data or model parameters special care needs to be taken to ensure that administrative units in the country and region masks match countries and/or regions used in the model or in other source datasets. For

example, the official release of LPJmL4, as described by Schaphoff et al. (2018b), distinguishes 236 countries and 242 regions ('Schaphoff18' in Table 4). While these numbers are close to the total number of countries and regions assigned here at 30' spatial resolution, GADM version 3.6 actually distinguishes fewer level-1 administrative units in Russia and more level-1 units in several of the other countries than are defined in LPJmL.

## 3.2 Soil

The aggregation methodology for soil data used in this toolbox does not derive 'average soils', but selects in each grid cell at the target resolution the 'dominant soil texture', which is derived by summing up the areas of all soil texture classes present in the source data and selecting the texture class with the largest area share. The most prominent soils in terms of global area share at both 5' and 30' spatial resolution are 'loam' and 'sandy loam' (Table 4). While the area shares covered by each soil texture class vary between the two datasets, the order from most prevalent to least prevalent is generally the same, except

for 'sandy clay', 'sily clay', and 'silty clay loam'. There are two reasons explaining the different relative prevalence of soil texture classes in the 5' and 30' dataset: 1) Both datasets use a slightly different land-sea mask. 2) Depending on the spatial heterogeneity of the source data, first aggregating HWSD data to 5' spatial resolution and then aggregating the 5' dataset to 30' spatial resolution is not expected to give the same result as aggregating HWSD data directly to 30' spatial resolution in all regions.





**Table 4.** Characteristics of the created grid, administrative and soil input datasets at 5' and 30' spatial resolution. Values from earlier input datasets used with LPJmL (e.g. in Schaphoff et al., 2018a) are provided for comparison (*Schaphoff18*).

| Parameter | 5' dataset | 30' dataset | Schaphoff18 dataset |
|---|---|---|---|
| **Land-sea mask, grid, country and region mask** | | | |
| Number of cells in grid | 2298226 | 67420 | 67420 |
| Total grid area [$10^6$ km$^2$] | 139.1 | 146.4 | 146.4 |
| Total land area [$10^6$ km$^2$] | 135.0 | 134.5* | 134.5* |
| Number of countries assigned (total: 257) | 254 | 237 | 236[†] |
| Number of cells assigned as 'No land' | 0 | 141 | 153[†] |
| Number of regions[‡] assigned (total: 252) | 252 | 242 | 242[†] |
| **Soil texture classes** (share of global grid cell area) | | | |
| % clay | 10.2 | 10.0 | 4.1 |
| % silty clay | 0.6 | 0.6 | 0.1 |
| % sandy clay | 0.7 | 0.6 | 0.9 |
| % clay loam | 5.0 | 4.5 | 14.1 |
| % silty clay loam | 0.7 | 0.6 | 0.3 |
| % sandy clay loam | 8.5 | 7.8 | 11.0 |
| % loam | 37.9 | 41.3 | 34.3 |
| % silt loam | 6.4 | 6.5 | 3.6 |
| % sandy loam | 12.4 | 11.7 | 24.2 |
| % silt | 0.0 | 0.0 | 0.0 |
| % loamy sand | 7.5 | 7.0 | 3.3 |
| % sand | 7.8 | 7.2 | 1.9 |
| % rock and ice | 2.4 | 2.2 | 2.0 |
| Number of cells gap-filled | 45865 | 753 | — |

* GADM-based land fractions in each cell are used, assigning a land fraction of 0 to cells without any land according to GADM.

[†] Country and region mapping in this dataset was not based on GADM version 3.6.

[‡] GADM level-1 units (provinces, states) are only distinguished in seven large countries: Australia, Brazil, Canada, China, India, Russia, USA



Roughly 2 % of the grid cells of the 5' grid and 1 % of the grid cells of the 30' grid fall outside the area covered by HWSD data and are gap-filled using the dominant soil texture in surrounding HWSD grid cells. The higher number of cells requiring gap-filling at 5' is to be expected since small data gaps in HWSD — e.g. missing soil information below inland water bodies — or small differences in the land-sea mask are less likely to cover full grid cells at 30' than at 5' spatial resolution.

    Finally, the last column in Table 4 compares the two datasets created using this toolbox to the *Schaphoff18* dataset. The latter

input used the same grid as the 30' version and the same version of HWSD source data. However, HWSD was aggregated to the target resolution by averaging the amount of sand, silt, and clay across cells at the source resolution and then deriving the texture class at the target resolution from the averaged shares (Schaphoff et al., 2013). The *Schaphoff18* dataset has a much higher share of grid cells with 'sandy loam' and 'clay loam' than the two datasets created with this toolbox. On the other hand, it has a much lower share of 'clay' and 'sand'. The comparison highlights the effect the choice of aggregation method has on

the derived dataset. None of the three datasets have any grid cells with assigned soil texture class 'silt'.

### 3.3 Hydrology

#### 3.3.1 River routing

River routing at 5' spatial resolution is based on HydroSHEDS version 1.1 (Lehner et al., 2008) and MERIT (Eilander et al., 2020), while we use DDM30 (Döll and Lehner, 2002) and STN-30 (Vörösmarty et al., 2000) at a spatial resolution of 30'.

Both DDM30 and STN-30 have been used in previous LPJmL applications. All drainage direction maps except STN-30 cover almost the full grid used here (Table 5). STN-30 has a smaller coverage of 93 % of the full grid area. Differences in coverage are mostly along coastlines. HydroSHEDS, MERIT, and DDM30 use a specific code for outlet cells. In addition, HydroSHEDS uses a specific code for inland sinks. STN-30 generally does not mark outlet cells. Any STN-30 cell that drains into a cell outside the covered land area is considered an outlet cell. Cells missing in the drainage direction maps are set to a missing

value in the generated drainage input file for LPJmL. They are treated like outlet cells in LPJmL.

    DDMs are a simplification of real-world river systems. Cells in a DDM may collect inflow from several upstream cells but drain all water into a single downstream cell. This poses limitations in representing river deltas or estuaries and can also shift or completely remove the location of watersheds between river basins. As a result, the spatial resolution of the DDM has implications on the total number and size of drainage basins as well as the location of the river outlet into the ocean, as shown

in Table 5. The median size of individual river basins at 30' resolution is more than 35 times the median size at 5'. The two 5' datasets are capable of distinguishing about 10 times more individual river basins consisting of at least 2 grid cells than the two 30' datasets. There are notable differences even for very large river basins, as shown for the Amazon, Congo, Mississippi, Nile, and Rio de la Plata Basin in Table 5. Differences between datasets at the same spatial resolution can be larger than the differences between the 5' and 30' datasets. For example, the area of the Rio de la Plata Basin is about 1/6 larger in both

STN-30 and MERIT than the respective basin areas in DDM30 and HydroSHEDS. The Nile Basin is about 1/4 to 1/3 larger in the STN-30 dataset than the other drainage direction maps. Outlet cell locations also differ by up to one degree in latitudinal and longitudinal direction between DDM30 and STN-30 (0.58 degree for HydroSHEDS and MERIT).



**Table 5.** Comparison of hydrological input datasets at 5' and 30' spatial resolution. 5' datasets are based on Lehner et al. (2008) (HydroSHEDS) and Eilander et al. (2020, 2021) (MERIT). 30' datasets are based on Döll and Lehner (2002) (DDM30) and Vörösmarty et al. (2000) (STN-30). Both the 5' and 30' lake and river mask are based on Lehner and Döll (2004). *Schaphoff18* refers to a previous input dataset for LPJmL (e.g. Schaphoff et al., 2018a).

| Parameter | 5' (HydroSHEDS) | 5' (MERIT) | 30' (DDM30) | 30' (STN-30) |
|---|---|---|---|---|
| **River routing** | | | | |
| % grid area covered by DDM | 99.5 | 99.9 | 98.4 | 92.6 |
| Median river basin size | 67 km$^2$ | 74 km$^2$ | 2643 km$^2$ | 2982 km$^2$ |
| River basins with at least 2 cells | 41780 | 42988 | 3900 | 3968 |
| Amazon Basin | $5.91 \cdot 10^6$ km$^2$ | $5.93 \cdot 10^6$ km$^2$ | $5.95 \cdot 10^6$ km$^2$ | $5.86 \cdot 10^6$ km$^2$ |
| | (51.04°W, 0.71°S) | (50.71°W, 0.46°S) | (50.25°W, 0.25°N) | (51.25°W, 0.75°S) |
| Congo Basin | $3.71 \cdot 10^6$ km$^2$ | $3.69 \cdot 10^6$ km$^2$ | $3.71 \cdot 10^6$ km$^2$ | $3.7 \cdot 10^6$ km$^2$ |
| | (12.46°E, 6.04°S) | (12.46°E, 6.04°S) | (12.25°E, 6.25°S) | (12.25°E, 5.75°S) |
| Mississippi Basin | $3.18 \cdot 10^6$ km$^2$ | $2.96 \cdot 10^6$ km$^2$ | $3.23 \cdot 10^6$ km$^2$ | $3.21 \cdot 10^6$ km$^2$ |
| | (89.71°W, 29.54°N) | (89.38°W, 28.96°N) | (89.25°W, 29.25°N) | (89.75°W, 29.25°N) |
| Nile Basin | $2.91 \cdot 10^6$ km$^2$ | $3.08 \cdot 10^6$ km$^2$ | $2.9 \cdot 10^6$ km$^2$ | $3.83 \cdot 10^6$ km$^2$ |
| | (30.38°E, 31.46°N) | (30.38°E, 31.46°N) | (31.25°E, 31.75°N) | (31.25°E, 31.75°N) |
| Rio de la Plata Basin | $2.59 \cdot 10^6$ km$^2$ | $2.99 \cdot 10^6$ km$^2$ | $2.58 \cdot 10^6$ km$^2$ | $3.02 \cdot 10^6$ km$^2$ |
| | (58.54°W, 33.96°S) | (58.29°W, 34.12°S) | (58.25°W, 34.25°S) | (58.25°W, 34.25°S) |
| **Irrigation** | | | | |
| Median upstream area of cell | 98 km$^2$ | 141 km$^2$ | 3192 km$^2$ | 3088 km$^2$ |
| Median upstream area of selected neighbour cell / median distance to neighbour cell / fraction of cells where selected neighbour has larger upstream area, depending on search algorithm | | | | |
| Adjacent | 1057 km$^2$ / 10 km / 99 % | 1156 km$^2$ / 10 km / 99 % | 23498 km$^2$ / 60 km / 96 % | 24693 km$^2$ / 60 km / 96 % |
| Adjacent (no upstream/downstream cell) | 365 km$^2$ / 10 km / 71 % | 407 km$^2$ / 10 km / 70 % | 10840 km$^2$ / 61 km / 73 % | 11469 km$^2$ / 60 km / 73 % |
| 75 km region (no upstream/downstream cell) | 14600 km$^2$ / 65 km / 96 % | 15175 km$^2$ / 66 km / 96 % | 10384 km$^2$ / 56 km / 72 % | 11105 km$^2$ / 56 km / 72 % |
| 75 km region (no u./d. cell, distance-weighted) | 7280 km$^2$ / 17 km / 87 % | 7593 km$^2$ / 17 km / 88 % | 9051 km$^2$ / 55 km / 57 % | 9237 km$^2$ / 55 km / 59 % |
| **Lake and river mask** | | | | |
| | 5' dataset | 30' dataset | Schaphoff18 dataset | |
| Global lake and river area | $2.783 \cdot 10^6$ km$^2$ | $2.476 \cdot 10^6$ km$^2$ | $2.456 \cdot 10^6$ km$^2$ | |



The spatial resolution also affects the irrigation scheme in LPJmL. The median upstream area per cell is more than 20–30 times larger at 30' compared to 5', reflecting the larger median river basin size at 30' (Table 5). We test four different options for selecting one neighbour cell from which to draw additional irrigation water: 1) directly adjacent cell with largest upstream area; 2) same as 1) but excluding upstream and downstream cells; 3) cell with largest upstream area in 75 km radius, excluding upstream and downstream cells; 4) same as 3) but applying an additional inverse distance weight with a power parameter of 2. Option 1) was used in previous LPJmL applications. The median upstream area of the neighbourhood cell under option 1) is 23498 and 24693 $\mathrm{km}^2$ at 30', and 1057 and 1156 $\mathrm{km}^2$ at 5' spatial resolution. As such, the median upstream area of the neighbour cell is about 7 to 10 times larger than the median upstream area of the cells themselves at 5' and 30'. The upstream area of the selected neighbour cell is larger than the upstream area of the cells themselves for more than 95 % of cells regardless of the spatial resolution. The median distance between each cell and its selected neighbour cell is 10 km at 5' and 60 km at 30', reflecting the different grid sizes of the datasets. Option 2) also selects the neighbour from directly adjacent cells but excludes upstream and downstream cells from the search because tapping the same river twice makes limited sense. This constraint leads to a substantial reduction of both the median upstream area of the neighbour cell as well as the fraction of all cells whose neighbour has a larger upstream area, while it has little effect on the median distance to the neighbour cell (Table 5). Apparently, option 1) is very likely to select a downstream cell as neighbour, which has a larger upstream area by definition but may not have additional water unless there is a confluence with another river. Option 3) selects the neighbour cell from all cells within a 75 km radius. For both 30' datasets, option 3) leads to similar results as option 2) since the 75 km search radius corresponds roughly to the distance between adjacent cells close to the equator. For the 5' datasets, expanding to a 75 km search radius increases the median upstream area of the neighbour cell and the distance to the neighbour cell to values that are even higher than in the 30' datasets. In fact, the distance to the selected neighbour cell appears to be close to the maximum search radius in many cells, which may not be realistic if there are closer cells with only a slightly smaller upstream area. Option 4) is a very simple attempt to provide a compromise between finding a neighbour cell with a larger upstream area while still limiting the transport distances. Adding an inverse distance weighting to the upstream cells of surrounding cells has relatively little effect for the datasets at 30' resolution: The median distance to the neighbour cell is reduced by only 1 km while the median upstream area of the neighbour cell is reduced by 13 to 17 %. At 5' spatial resolution, option 4) reduces the median distance to the selected neighbour cell by almost 75 % while the median upstream area is only reduced by roughly 50 %. The majority of cells are still assigned a neighbour cell with an upstream area more than ten times their own. Toolbox users may try out further values for the power parameter but a comprehensive assessment of the best parameter combination would require simulations with LPJmL to test the effect of different neighbour irrigation inputs on simulated irrigation water withdrawals and irrigated crop yields.

### 3.3.2 Lakes and rivers

The bottom row of Table 5 compares the global area covered by lakes and rivers at the two spatial resolutions. Both datasets are based on aggregating gridded GLWD level 3 data to the target grid. The 5' dataset has about 12 % more area covered by lakes and rivers than the 30' dataset. This difference is mainly due to the Caspian Sea, which is included in the 5' grid but





excluded from the 30' grid. It should be noted that lake and river fractions extracted from GLWD level 3 differ slightly from those extracted by polygon intersection from level 1 and 2 of the source data. However, we find that the difference is less than $400 \, \mathrm{km}^2$ ($< 0.02$ %) at the global scale.

765    The last column of the table provides the lake and river area from the Schaphoff18 input dataset which used the same grid as the 30' dataset created by this toolbox but saved grid cell fractions covered by lakes and rivers with a reduced accuracy of integer percent values. For further comparison, we note that the HydroLAKES dataset reports a global surface area of natural lakes of $2.67 \cdot 10^6 \, \mathrm{km}^2$, which is about 4 % smaller than the area covered by lakes and rivers in our 5' dataset or about 8 % larger than our 30' dataset (Messager et al., 2016). HydroLAKES does not include areas covered by rivers.

770    ### 3.3.3    Dams and reservoirs

GRanD version 1.3 (Lehner et al., 2011) is used for this toolbox application. It includes entries for 7320 dams and reservoirs. Before assigning dams to grid cells, 40 GRanD records with missing storage capacity or marked as either 'Destroyed', 'Planned', 'Removed', 'Replaced', 'Subsumed', or 'Under construction' are removed. Missing reservoir areas for 14 reservoirs are estimated using average relationships between area and storage capacity (Lehner et al., 2011). No dams are outside the area 775    covered by either the 5' or 30' grid. We set a maximum search radius of 1 degree for both resolutions, but enforce a maximum distance between original dam location and assigned grid cell center of 15 km and 75 km at 5' and 30', respectively. Values of 1 and 1.5 are used for distance penalty $p_{\mathrm{dis}}$ and deviation penalty $p_{\mathrm{dev}}$, testing permutations where both penalties have the same value or one is higher than the other. No $p_{\mathrm{sign}}$ is used to assign a different penalty based on whether the catchment area is over or underestimated. The number of cells with assigned reservoirs is lower than the total number of reservoirs in GRanD 780    since dams and reservoirs are sometimes located close to each other. The 7280 dams are assigned to 7040–7064 individual grid-cells at 5' resolution, depending on the DDM and assignment strategy (Table 6). The number of grid-cells with dams is reduced further to 4077–4579 at 30' spatial resolution. The median distance between reported and assigned dam location is about 3 and 20 km at 5' and 30', respectively, if dams are simply assigned to the grid-cell their reported coordinates fall into ('grid-based' in Table 6). Distances between reported and assigned locations increase for all weighting-based assignment 785    strategies. This increase is smallest for the strategy with a higher distance weight ($p_{\mathrm{dis}} = 1.5, p_{\mathrm{dev}} = 1$ in Table 6) and highest for the strategy with a higher deviation weight ($p_{\mathrm{dis}} = 1, p_{\mathrm{dev}} = 1.5$). While absolute distances are smaller at 5' than at 30', the relative increase of weighting-based assignment compared to the grid-based assignment is larger at 5'. The weighting-based assignment is less likely to differ from the grid-based assignment at 30' than at 5' (see '% of dams not grid-based' in Table 6). The main reason for using a weighting-based assignment strategy is to achieve a better match between catchment areas re-790    ported in GRanD and DDM-based upstream areas of the assigned cells, which is used as an indication that modelled dams are assigned to the correct river basin. DDM-derived upstream areas are commonly larger than reported catchment areas, as evidenced by the much higher percentages of dams with overestimation compared to dams with underestimation in Table 6. This is especially true for the datasets at 30', where reported catchment areas are often smaller than one grid-cell. The percentage numbers in brackets account for this by setting reported catchment areas that are smaller than one grid-cell to the size of





**Table 6.** Comparison of reservoir input datasets at 5' and 30' spatial resolution. 5' datasets are produced for DDMs by Lehner et al. (2008) (HydroSHEDS) and Eilander et al. (2020, 2021) (MERIT). 30' datasets are produced for DDMs by Döll and Lehner (2002) (DDM30) and Vörösmarty et al. (2000) (STN-30). Reported catchment areas may be smaller than a single cell in the DDM. Numbers in brackets set reported catchment areas smaller than one grid-cell to the size of the grid-cell before comparison with DDM upstream areas.

| Parameter | Assignment strategy | 5' (HydroSHEDS) | 5' (MERIT) | 30' (DDM30) | 30' (STN-30) |
|---|---|---|---|---|---|
| Number of cells with reservoirs | grid-based | 7051 | 7051 | 4579 | 4579 |
| | $p_{\text{dis}} = 1, p_{\text{dev}} = 1$ | 7045 | 7049 | 4198 | 4263 |
| | $p_{\text{dis}} = 1.5, p_{\text{dev}} = 1$ | 7061 | 7064 | 4325 | 4353 |
| | $p_{\text{dis}} = 1, p_{\text{dev}} = 1.5$ | 7040 | 7045 | 4077 | 4171 |
| Median distance [m] | grid-based | 3272 | 3272 | 19580 | 19580 |
| | $p_{\text{dis}} = 1, p_{\text{dev}} = 1$ | 4310 | 4418 | 24317 | 23763 |
| | $p_{\text{dis}} = 1.5, p_{\text{dev}} = 1$ | 3882 | 4002 | 22510 | 22241 |
| | $p_{\text{dis}} = 1, p_{\text{dev}} = 1.5$ | 4715 | 4836 | 26097 | 25454 |
| % of dams not grid-based | grid-based | 0 | 0 | 0 | 0 |
| | $p_{\text{dis}} = 1, p_{\text{dev}} = 1$ | 43.9 | 45.4 | 37.1 | 34.7 |
| | $p_{\text{dis}} = 1.5, p_{\text{dev}} = 1$ | 36.6 | 38.7 | 30.9 | 28.9 |
| | $p_{\text{dis}} = 1, p_{\text{dev}} = 1.5$ | 49.7 | 51.3 | 42.3 | 39.9 |
| % of dams strong underestimation | grid-based | 5.6 | 4.8 | 1.4 | 3.8 |
| | $p_{\text{dis}} = 1, p_{\text{dev}} = 1$ | 0.7 | 0.5 | 0.2 | 1.3 |
| | $p_{\text{dis}} = 1.5, p_{\text{dev}} = 1$ | 1 | 0.6 | 0.3 | 1.6 |
| | $p_{\text{dis}} = 1, p_{\text{dev}} = 1.5$ | 0.5 | 0.3 | 0.2 | 1.1 |
| % of dams medium underestimation | grid-based | 6.6 | 5.1 | 2 | 4.5 |
| | $p_{\text{dis}} = 1, p_{\text{dev}} = 1$ | 11 | 9.4 | 4.8 | 6.8 |
| | $p_{\text{dis}} = 1.5, p_{\text{dev}} = 1$ | 10.3 | 8.5 | 4.2 | 6.4 |
| | $p_{\text{dis}} = 1, p_{\text{dev}} = 1.5$ | 12.2 | 10.4 | 5.2 | 7.3 |
| % of dams good match | grid-based | 34.4 (34.5) | 35.2 (35.3) | 8.1 (8.1) | 5.5 (5.5) |
| | $p_{\text{dis}} = 1, p_{\text{dev}} = 1$ | 52.3 (52.4) | 54.1 (54.2) | 14.9 (15) | 11.6 (11.7) |
| | $p_{\text{dis}} = 1.5, p_{\text{dev}} = 1$ | 49.9 (50) | 52 (52.2) | 14.2 (14.2) | 11 (11.1) |
| | $p_{\text{dis}} = 1, p_{\text{dev}} = 1.5$ | 53.6 (53.7) | 55.5 (55.6) | 15.3 (15.3) | 12 (12.1) |
| % of dams medium overestimation | grid-based | 28.1 (34.9) | 29.1 (35.1) | 13.2 (53.3) | 12 (52.8) |
| | $p_{\text{dis}} = 1, p_{\text{dev}} = 1$ | 22.8 (34) | 22.9 (33.7) | 13.2 (71.8) | 13.5 (72.5) |
| | $p_{\text{dis}} = 1.5, p_{\text{dev}} = 1$ | 24.7 (35.1) | 24.7 (34.4) | 13.7 (68.5) | 13.3 (68.6) |
| | $p_{\text{dis}} = 1, p_{\text{dev}} = 1.5$ | 21 (32.7) | 21 (32.6) | 12.9 (74.9) | 13.1 (75.3) |
| % of dams strong overestimation | grid-based | 25.4 (18.5) | 25.8 (19.7) | 75.2 (35.1) | 74.3 (33.4) |
| | $p_{\text{dis}} = 1, p_{\text{dev}} = 1$ | 13.2 (1.9) | 13.2 (2.2) | 66.8 (8.1) | 66.8 (7.7) |
| | $p_{\text{dis}} = 1.5, p_{\text{dev}} = 1$ | 14.1 (3.6) | 14.1 (4.3) | 67.7 (12.8) | 67.7 (12.3) |
| | $p_{\text{dis}} = 1, p_{\text{dev}} = 1.5$ | 12.8 (0.9) | 12.8 (1.1) | 66.4 (4.4) | 66.5 (4.1) |

strong underestimation: DDM upstream area < 50 % of reported catchment area; medium underestimation: DDM upstream area $\geq$ 50 % and $\leq$ 90 % of reported catchment area; good match: DDM upstream area within $\pm$10 % of reported catchment area; medium overestimation: DDM upstream area > 10 % and $\leq$ 100 % above reported catchment area; strong overestimation: DDM upstream area larger than twice the reported catchment area.





795 the grid-cell before comparing them to DDM-derived upstream areas. This alternative evaluation shifts the majority of dams having a 'strong overestimation' of DDM-derived upstream areas to the 'medium overestimation' category.

Overall, reported catchment areas cannot be matched well at 30' spatial resolution. Using grid-based assignment, only 5.5–8 % of dams have a DDM-derived upstream area within 10% of the reported catchment area ('good match' in Table 6). This increases to 11–15 % of dams using the weighting-based assignment strategies. Yet, the upstream areas of about 2/3 of all dams 800 are still more than twice the reported catchment area even with weighting-based assignment strategies.

Reported catchment areas can be matched much better at 5' spatial resolution. More than half of all dams show a 'good match' using the weighting-based assignment strategies, while only 13–15 % of dams show a strong over/underestimation.

The weighting-based assignment strategies generally shift dams from the more extreme categories (e.g. 'strong over/underestimation') towards a better match between DDM-derived upstream area and reported catchment area at both spatial resolutions, yet they 805 are much more effective at 5' than at 30'. While upstream areas of many reservoirs are still overestimated using weighting-based assignment, especially at 30', we note that this problem relates mostly to small reservoirs with small catchment areas. The weighting-based assignment strategies greatly improve positioning of the major reservoirs (Figure S1 in the supporting information). As expected, the weighting-based approach with $p_{\mathrm{dev}} = 1.5$ provides the best match between DDM and reported areas, but it also leads to larger distances between reported and assigned positions than the other assignment approaches. This 810 increases the risk that dams are assigned to the wrong river (even though the upstream area shows a better match) or that upstream-downstream relationships in regions with multiple dams in close vicinity are not represented correctly. While using a larger maximum search distance than in our test application may improve the match between DDM-derived upstream areas and reported catchment areas it would also increase the risk of a mis-representation of the river network topology.

The assignment algorithms implemented in the toolbox cannot prevent such mis-assignments, but they provide ample diag-815 nostics information, including mapping reported and assigned dam locations to allow for manual visual inspection.

### 3.4 Land use and land management

### 3.4.1 Land use

In this section we first describe the land use datasets created with this toolbox and then compare our datasets to another land use dataset that was recently developed for phase 2b of the Inter-Sectoral Impact Model Intercomparison Project (ISIMIP2b, 820 Frieler et al., 2017) to highlight the effect of some of the assumptions used in generating these kinds of datasets. We use FAOSTAT data downloaded in January 2020 and AQUASTAT data downloaded in April 2020 for this toolbox application. Since FAOSTAT and AQUASTAT source data are updated frequently by the FAO, we expect that potential users of this toolbox will not use the same version of the data, so their results will likely differ somewhat from ours. All data integration of country-level and gridded source data as well as the spatial disaggregation of country-level crop-specific harvested areas to the grid 825 level is carried out at 5' spatial resolution. The land use datasets created by the toolbox cover the period 1500–2017. Figure 2 shows the evolution of global cropland extent and harvested areas over that period. Global total cropland and irrigated cropland expanded from $2.6 \cdot 10^6$ km$^2$ and 42009 km$^2$ in 1500 to $16 \cdot 10^6$ km$^2$ and $2.8 \cdot 10^6$ km$^2$ in 2017, respectively. Total harvested

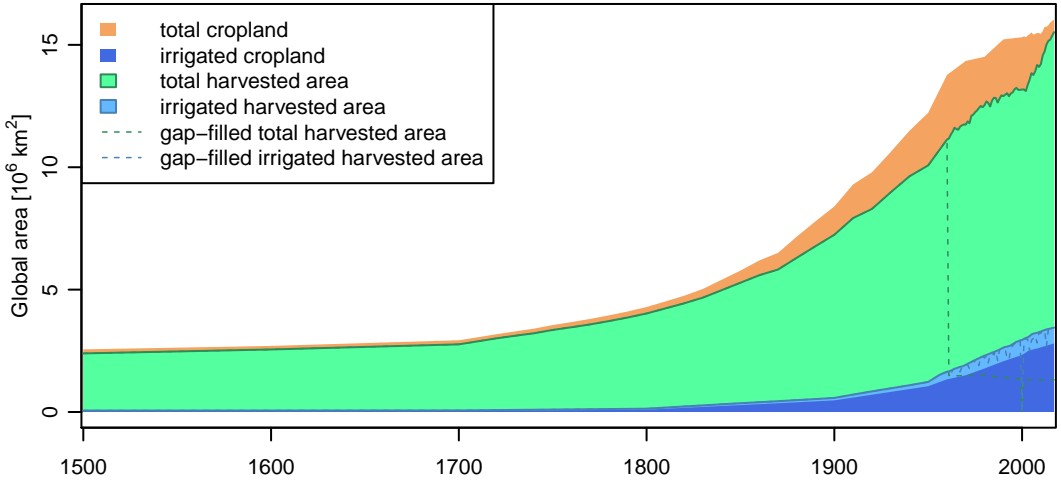

**Figure 2.** Evolution of global cropland extent and harvested areas over time. Cropland extent is based on HYDE version 3.2.1 (Klein Gold-ewijk et al., 2017). Harvested areas are based on a combination of data from AQUASTAT (FAO, 2020a), FAOSTAT (FAO, 2020b), MIRCA2000 (Portmann et al., 2010), and Monfreda et al. (2008). Dashed lines indicate the part of total and irrigated harvested areas that are based on gap-filling or extrapolation of country-level source data. 100 % of total harvested areas are based on extrapolation before the start of FAOSTAT data in 1961; gap-filled total harvested areas decrease between 1961 and 2017 with improving data availability in FAOSTAT. Source datasets for irrigated harvested areas cover only individual years, not continuous time series.

areas were smaller than total cropland extent over the whole period, expanding from $2.4 \cdot 10^6$ km$^2$ in 1500 to $15.5 \cdot 10^6$ km$^2$ in 2017. Irrigated harvested areas were larger than irrigated cropland extent over the whole period, expanding from 58844 km$^2$ in 830    1500 to $3.4 \cdot 10^6$ km$^2$ in 2017. The source datasets used to derive crop-specific harvested areas only cover the year 2000 in the case of MIRCA2000 and the MON dataset, 1961–2016 in the case of AQUASTAT, and 1961–2017 in the case of FAOSTAT. FAOSTAT data are not complete for all countries and all years but more than 90 % of all crop-country data series contain more than 20 data points. Data availability for irrigated harvested areas is generally much lower than for total crop-specific harvested areas. Even though AQUASTAT technically covers almost the same time period as FAOSTAT only about one third 835    of all crop-country data series in AQUASTAT contain more than one data point, only about 10 % contain more than two data points. As such, extrapolation and gap-filling are used to expand the source data to the full period of 1500–2017. The extent of total harvested areas based on gap-filling decreases from roughly $1.5 \cdot 10^6$ km$^2$ (13 %) to $1.3 \cdot 10^6$km$^2$ (8 %) over the period covered by FAOSTAT data, shown as a dashed green line in Figure 2. This includes crops not included in FAOSTAT but added to the dataset based on data from the MON dataset. Due to the sparsity of source data on irrigated harvested areas, there are 840    only 12 years in the whole time series where less than 90 % of global irrigated harvested areas are based on extrapolation or gap-filling of MIRCA2000 and AQUASTAT, shown as a dashed blue line in Figure 2. The year 2000, which is covered by all four source datasets, requires the least amount of gap-filling with less than 0.2 % of total harvested areas and less than 0.02 % of irrigated harvested areas. Before 1961, all harvested areas are based on extrapolation.





**Table 7.** Comparison of crop-specific harvested areas and assigned growing areas [in $\text{km}^2$] in created land use datasets

| Crop | Harvested area | | 5' dataset | | 30' dataset | | ISIMIP dataset | |
|---|---|---|---|---|---|---|---|---|
| | 2000 | 2015 | 2000 | 2015 | 2000 | 2015 | 2000 | 2015 |
| rainfed temperate cereals | 2106308 | 1991616 | 2063663 | 1935294 | 2067463 | 1940179 | 2937016 | 2810085 |
| rainfed rice | 600774 | 565497 | 483448 | 402082 | 489111 | 407421 | 841796 | 905655 |
| rainfed maize | 1228869 | 1681421 | 1137057 | 1425513 | 1151622 | 1441479 | 1583874 | 1612329 |
| rainfed tropical cereals | 745072 | 676608 | 716802 | 634586 | 719632 | 636733 | 845296 | 1056260 |
| rainfed pulses | 643879 | 792798 | 594350 | 690432 | 600220 | 693288 | 672905 | 737776 |
| rainfed temperate roots | 228529 | 200088 | 211031 | 164096 | 211800 | 164164 | 289672 | 276691 |
| rainfed tropical roots | 316777 | 435023 | 257305 | 317260 | 262040 | 322557 | 275695 | 352874 |
| rainfed oil crops sunflower | 199582 | 235378 | 192850 | 228596 | 193337 | 228852 | 229185 | 240011 |
| rainfed oil crops soybean | 683907 | 1119022 | 627748 | 892928 | 636437 | 914150 | 710160 | 761091 |
| rainfed oil crops groundnut | 195717 | 216949 | 178579 | 188881 | 179442 | 189858 | 219607 | 259431 |
| rainfed oil crops rapeseed | 223544 | 303889 | 196792 | 258237 | 197677 | 258306 | 214857 | 196384 |
| rainfed sugar cane | 93505 | 141751 | 84282 | 118469 | 86481 | 122949 | 184401 | 198473 |
| rainfed other crops | 2969135 | 3447805 | 2723766 | 2939644 | 2736159 | 2946060 | 3683564 | 3729802 |
| rainfed pasture/managed grass[†] | 33226773 | 32413397 | 33226773 | 32413397 | 33214257 | 32401395 | 33216990 | 32819305 |
| rainfed bio-energy grass | 0 | 0 | 0 | 0 | 0 | 0 | 0 | 0 |
| rainfed bio-energy tree | 0 | 0 | 0 | 0 | 0 | 0 | 0 | 0 |
| irrigated temperate cereals | 715603 | 832141 | 478935 | 575336 | 483054 | 581307 | 526978 | 580714 |
| irrigated rice | 939250 | 1058303 | 584772 | 666232 | 594531 | 676376 | 454827 | 526886 |
| irrigated maize | 293810 | 374422 | 216182 | 281259 | 217527 | 283614 | 221285 | 267953 |
| irrigated tropical cereals | 51647 | 57671 | 41123 | 44567 | 41476 | 44807 | 98066 | 112952 |
| irrigated pulses | 60731 | 70390 | 50101 | 58502 | 50265 | 58713 | 127564 | 147174 |
| irrigated temperate roots | 52798 | 60348 | 43085 | 50322 | 43289 | 50543 | 42960 | 50085 |
| irrigated tropical roots | 7509 | 8563 | 5324 | 6138 | 5341 | 6177 | 24169 | 30483 |
| irrigated oil crops sunflower | 11999 | 19309 | 10384 | 16151 | 10451 | 16390 | 18840 | 20161 |
| irrigated oil crops soybean | 59939 | 88987 | 45839 | 67970 | 46147 | 68504 | 82773 | 97371 |
| irrigated oil crops groundnut | 36345 | 47999 | 25853 | 33667 | 25973 | 33898 | 37218 | 43568 |
| irrigated oil crops rapeseed | 34038 | 39051 | 26591 | 30429 | 26671 | 30527 | 52020 | 60302 |
| irrigated sugar cane | 100258 | 124748 | 68773 | 85119 | 69343 | 86379 | 50148 | 66659 |
| irrigated other crops | 575204 | 624579 | 459374 | 501378 | 463682 | 506492 | 626401 | 701180 |
| irrigated pasture/managed grass[†] | 0 | 0 | 0 | 0 | 0 | 0 | 0 | 0 |
| irrigated bio-energy grass | 0 | 0 | 0 | 0 | 0 | 0 | 0 | 0 |
| irrigated bio-energy tree | 0 | 0 | 0 | 0 | 0 | 0 | 0 | 0 |
| rainfed crop sum | 10235598 | 11807843 | 9467673 | 10196017 | 9531421 | 10265996 | 12688027 | 13136863 |
| irrigated crop sum | 2939130 | 3406511 | 2056336 | 2417069 | 2077749 | 2443727 | 2363249 | 2705487 |
| total crop sum | 13174727 | 15214354 | 11524009 | 12613087 | 11609170 | 12709723 | 15051276 | 15842350 |
| rainfed fallow[‡] | n.a. | n.a. | 3514082 | 2962020 | 3444582 | 2885480 | 0 | 0 |
| irrigated fallow[‡] | n.a. | n.a. | 274120 | 338554 | 252163 | 311095 | 0 | 0 |
| total crops + fallow | n.a. | n.a. | 15312210 | 15913660 | 15305915 | 15906298 | 15051276 | 15842350 |

[†] pasture/managed grass not included in rainfed/irrigated/total crop sums

[‡] Fallow land is not applicable (n.a.) to the harvested area dataset. It is derived from the difference between available cropland and assigned crop growing areas.



Table 7 disaggregates global total sums over all crops shown in Figure 2 and provides global totals for the CFTs simu-
lated by LPJmL for the years 2000 and 2015. The 'Harvested area' columns are consistent with Figure 2. As mentioned in
Section 2.5.5, LPJmL currently does not support simulating multiple cropping systems. Therefore, harvested areas exceeding
physical cropland in individual cells are scaled down to crop-specific growing areas that fit into physical cropland for the final
land use input datasets. For the 30' dataset, crop-specific harvested areas and physical cropland are aggregated from 5' to 30'
before scaling down any harvested areas exceeding physical cropland in individual cells at 30'. Table 7 gives global sums of the
crop-specific growing areas at 5' and 30' in the columns labelled '5' dataset' and '30' dataset'. The columns labelled 'ISIMIP
dataset' compare crop-specific growing areas based on this toolbox to the ISIMIP2b dataset. The ISIMIP2b dataset uses the
same grid as the 30' dataset from this toolbox.

The global sum of harvested areas across all crops increases from $13.2 \cdot 10^6$ km$^2$ in 2000 to $15.2 \cdot 10^6$ km$^2$ in 2015, of which
roughly 22 % are irrigated and 78 % are rainfed ('total crop sum' and 'irrigated crop sum' in Table 7). Assigned growing areas
for all crops are lower than the respective harvested areas in both the 5' dataset and 30' dataset. Aggregated across all crops,
assigned growing areas are roughly 11.9–12.5 % smaller than harvested areas in 2000 and 16.5–17.9 % smaller than harvested
areas in 2015. This suggests that the practice of multiple cropping has expanded over that period. We note that the difference
between harvested areas and growing areas does not correspond directly to the crop area that is actually grown in multiple
cropping systems since multiple cropping and fallow land — i.e. cropland not currently used to grow any crops — can occur
within the same grid cell but are not explicitly accounted for in the toolbox. Also, the toolbox does not distinguish between
annual and perennial crops in its processing of harvested areas. Assigned growing areas are slightly larger in the 30' dataset
than the 5' dataset. This is an artefact of the aggregation from 5' to 30' in which some 5' grid cells with multiple cropping
and other grid cells with fallow land may average out at the 30' scale. The toolbox designates any cropland area that exceeds
the sum of crop-specific growing areas in a grid cell as fallow land. In Table 7 the sum of 'total crops + fallow' corresponds
to the total available cropland area. We note that this approach differs from other land use datasets. For example, Hurtt et al.
(2020) disaggregate cropland into five crop types assuming that each crop's relative share of cropland is equal to its relative
share of total harvested areas. The Hurtt et al. (2020) dataset provides the basis of the ISIMIP2b dataset used as comparison
here (Frieler et al., 2017). Another previous version of a land use input dataset for LPJmL assigned cropland without harvested
areas to the 'others' category (Fader et al., 2010). As mentioned before, multiple cropping and fallow land can occur within
the same grid cell in the real world so this approach is likely to underestimate fallow land. Globally, fallow land shrinks from
$3.7$–$3.8 \cdot 10^6$ km$^2$ in 2000 to $3.2$–$3.3 \cdot 10^6$ km$^2$ in 2015 while total cropland increases from $15.3 \cdot 10^6$ km$^2$ to $15.9 \cdot 10^6$ km$^2$
over the same period, again suggesting an overall increase in cropping intensity. While the toolbox allows to create an input
dataset of grid-level irrigated/rainfed fallow land, fallow land is currently not implemented as a land use category in LPJmL
(but planned for future model versions). Using the land use dataset created by this toolbox effectively reduces global cropland
compared to other land use input datasets.

Even though the total grid area of the 30' dataset exceeds that of the 5' dataset by $7.3 \cdot 10^6$ km$^2$ (Table 4), this mismatch
of the land-sea mask has almost no effect on the land use dataset, with a difference in global cropland area of just 6295 km$^2$





(0.04 %) in 2000 and 7362 km$^2$ in 2015. Likewise, global pasture/managed grassland areas in the two datasets differ by just 12516 km$^2$ (0.04 %) and 12002 km$^2$ in 2000 and 2015.

Breaking down the total sums into individual crops, we note that the difference between assigned growing areas and harvested areas varies widely among crops but is generally larger for irrigated crops than for rainfed crops, suggesting that multiple cropping is more prominent in irrigated crops. Due to the missing representation of multiple cropping in the assigned growing areas, crop-specific irrigation shares (irrigated crop area as a fraction of total crop area) are generally lower in the 5' and 30' dataset than in the harvested area dataset. Still, the ranking of crops from most to least irrigated is mostly kept intact except for
temperate roots and maize which are swapped in 2000, and temperate cereals, temperate roots, soybean, and tropical cereals in 2015. The former have almost identical irrigation shares, while irrigated temperate roots are the only crop group actually having a larger irrigation share in our 5' and 30' dataset than in the harvested area dataset for the year 2015.

To our knowledge, our land use datasets at 5' and 30' are the only time series of crop-specific growing areas that take into account changes of the relative crop shares over time. Other comparable datasets like the ISIMIP2b dataset or previous datasets
developed for LPJmL (e.g. Bondeau et al., 2007; Fader et al., 2010; Jägermeyr et al., 2015) use static crop shares in each grid cell and only scale cropland extent over time. As such, we expect that our land use datasets should perform better in capturing trends of crop-specific areas over time. Comparing crop-specific harvested areas in 2000 and 2015, four rainfed CFTs show a decreasing trend while the remaining rainfed crops and all irrigated crops show an increasing trend. Our growing area datasets at 5' and 30' resolution capture the direction of change for all crops. For rainfed crops with a negative trend in harvested areas,
the trend is even more negative in our growing area datasets, whereas positive trends for rainfed CFTs are generally smaller in our datasets than the observed trends in harvested areas, with the exception of rainfed sunflower. This dampening of positive trends and amplification of negative trends is consistent with an increase in rainfed cropping intensity over that period, i.e. more rainfed crops grown in multiple cropping systems. For irrigated crops, the trends in our datasets are slightly amplified compared to the corresponding harvested area trends for half of the CFTs and dampened for the other half. Aggregated across
all irrigated crops, irrigated growing areas in our 5' and 30' dataset show a more positive trend than irrigated harvested areas. At the same time, irrigated fallow land also increases, suggesting that irrigated cropland expands at a faster pace than irrigated harvested areas, leading to a decrease in irrigated cropping intensity.

Total cropland in the ISIMIP2b dataset is very similar to our 30' dataset. While crop shares evolve over time in our dataset, the ISIMIP2b dataset uses constant crop shares representative of the year 2000 and only scales cropland extent over time
to derive its time series. Similarly to our dataset, the ISIMIP2b dataset scales down harvested areas whenever they exceed available cropland in a grid cell. However, harvested areas are also scaled up to fill all available cropland whenever available cropland exceeds harvested areas. As a result, the ISIMIP2b dataset has much larger growing areas and no fallow land. Finally, while our dataset splits total crop-specific harvested areas for each crop into rainfed and irrigated harvested areas based on data from MIRCA2000 and AQUASTAT, the ISIMIP2b dataset uses a very simple approach and assigns the same irrigated share
to all crops present in a grid cell. These differences in methodology lead to substantial differences between both datasets even for the year 2000. Crop-specific growing areas (rainfed and irrigated combined) of all crops are larger in the ISIMIP2b dataset than in our dataset. They even exceed crop-specific harvested areas for all crops except rice. Irrigation shares per crop are





underestimated by up to 60 % for heavily irrigated crops (e.g. sugar cane, rice, temperate cereals, maize) and overestimated up to a factor of 3.5 for less irrigated crops (e.g. tropical roots, pulses, tropical cereals). Since the ISIMIP2b dataset uses constant

crop shares in each grid cell it also performs poorly in capturing crop-specific trends over time. Rainfed soybeans, sugar cane and maize show strong positive trends in harvested areas, but show little trend in the ISIMIP2b data. Rainfed rapeseed, which shows a 36 % increase in harvested areas, even shows a 9 % decrease in the ISIMIP2b dataset. On the other hand, rainfed rice and tropical cereals show a decreasing trend in harvested areas but an increasing trend in the ISIMIP2b dataset. While all datasets agree on increasing trends for all irrigated crops, trends in our growing area datasets are often closer to the harvested

area dataset than the trends in the ISIMIP2b dataset.

The description in this section focuses only on global sums of crop-specific growing areas and does not look at differences between the datasets at the country scale or for even smaller regions. Overall, we believe that our new datasets capture trends in crop-specific growing areas better than previous datasets, at least from the global down to the country scale. Spatial crop patterns within countries are highly uncertain, as has been emphasized both by data publications describing such gridded crop-

specific datasets as well as publications analysing and comparing available datasets (e.g. Leff et al., 2004; You and Wood, 2006; Monfreda et al., 2008; Portmann et al., 2010; Anderson et al., 2015; Joglekar et al., 2019; Yu et al., 2020; Grogan et al., 2022). Spatial patterns of the datasets created with this toolbox are determined to a large extent by the patterns from Monfreda et al. (2008). As pointed out in the documentation accompanying the MON data download: "These data sets are not intended to be interpreted at the individual grid cell level. Although it is presented at five-minute resolution, the best way

to use these maps is to compare counties, states, regions, countries, or continents" (Earthstat team, 2020). The same holds true for the land use patterns from this toolbox. The toolbox itself does not check whether crops are assigned to locations suitable for their cultivation. The implicit assumption made in the toolbox is that the spatial patterns from the MON dataset (or any alternative dataset used to replace it) can be used to derive suitable growing areas. However, the spatial redistribution algorithm (Section 2.5.4) may assign some harvested areas outside of suitable regions if they cannot be fit into the existing

cropland within growing areas defined by the MON dataset. The toolbox provides extensive diagnostic outputs in addition to the generated harvested area pattern. Table S1 in the supporting online information gives an example of these diagnostics for the spatial disaggregation of harvested areas in Burkina Faso in 2015. Column 'redist.' refers to the harvested area relocated by the spatial redistribution algorithm. A high value of redistributed harvested area indicates that harvested areas calculated by Equation 12 have overshot multiple cropping suitability in many cells of the country. This can be caused by a number of

reasons such as: 1) changes in the relative importance of crops within a country over time, e.g. an expansion of cash crops or energy crops compared to staple food crops, 2) uneven cropland expansion or abandonment in some parts of the country, 3) inconsistencies between the year-2000 crop-specific areas in the MON dataset and FAOSTAT. Column 'additive' refers to harvested areas that could not be assigned successfully by the iterative redistribution algorithm within the maximum number of 1000 iterations. They are assigned to grid-cells using a simple additive approach (Section 2.5.4). Column 'OOP' refers to

harvested areas assigned to grid cells out of pattern, i.e. grid cells where the crop is not grown according to the MON dataset. Harvested areas out of pattern may put crops into regions that are climatically unsuitable. In some cases, up to 100 % of total harvested areas of a crop can be assigned out of pattern. There are two main reasons for this: 1) the MON dataset does not

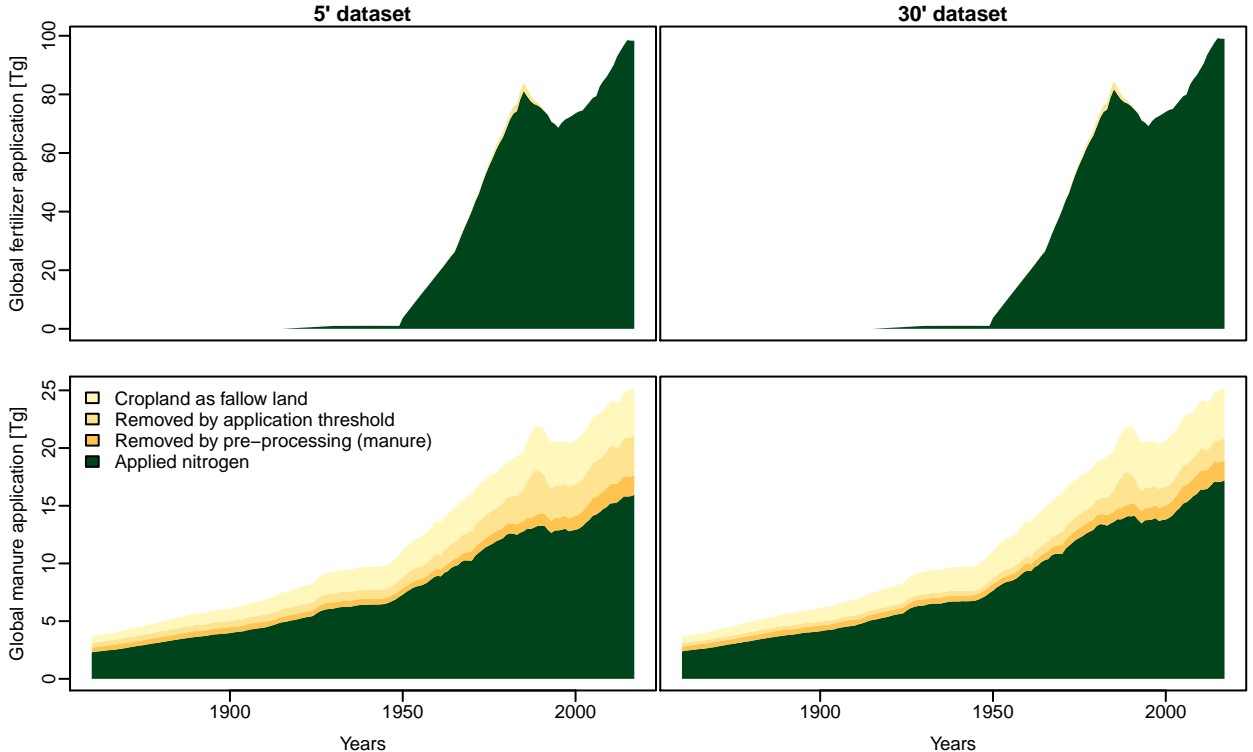

**Figure 3.** Evolution of global mineral fertilizer and manure application over time.

have a pattern for that crop in that country, indicated by 'Pattern = FALSE' in the last column of Table S1, 2) problems caused

by the temporal extrapolation of country-level harvested areas before the period covered by FAOSTAT data. The US is a good

example of the latter: Major present-day crop growing regions in the US had basically no cropland just 200–250 years ago.

Since the dataset created by this toolbox covers the period 1500–2017, present-day crop patterns from the MON dataset may

lie outside historical cropland areas.

### 3.4.2 Nitrogen fertilizer and manure

Application of mineral fertilizers in the generated datasets first starts in 1916 with a global sum of 0.06 Teragrams (Tg).

Fertilizer application increases rapidly from 1 Tg in 1949 to 81–82 Tg in 1985, then decreases to roughly 69 Tg in 1995

before rising again to 98–99 Tg in 2017 (top row Figure 3). Total application is slightly higher in the 30' dataset than than

5' dataset, although differences are less than 1 %. A maximum application threshold of 500 kg/ha is applied to fertilizer

rates in the toolbox, which reduces total annual fertilizer application by up to 2.9 Tg (3.4 %) and 2.7 Tg (3.2 %) at 5' and

30', respectively ('Removed by application threshold' in Figure 3). The reduction is slightly lower in the 30' dataset, because

the maximum application rate is enforced at the target resolution, where some extreme values may average out in the spatial

aggregation.





Global manure application to crops increases from 2.3–2.4 Tg in 1860 to 15.9–17.2 Tg in 2017 in the generated datasets (bottom row in Figure 3). Global manure application is between 4–8 % higher in the 30' dataset than the 5' dataset. However, manure application is substantially lower than the global sums reported in the source dataset (Zhang et al., 2017) at both resolutions. A total of 4–12 % of annual manure applied to croplands in the source dataset is lost when rescaling application rates 'per grid' to rates 'per cropland' because there is no cropland according to the HYDE dataset in several grid cells, where the source dataset (Zhang et al., 2017) reports manure application (labeled as 'Removed by pre-processing' in Figure 3). Similar to the fertilizer dataset, a maximum application threshold of 100 kg/ha is enforced at the target resolution, which cuts off about 8–18 % (4–13 %) of the total manure amount from the source dataset at 5' (30') spatial resolution. Finally, the treatment of fallow land — which is a part of total cropland — has large implications on the total amount of applied manure. Assuming no manure application on fallow land, an additional 15–21 % (16–22 %) of the total annual manure amount from the source dataset is lost at 5' (30') ('Cropland as fallow land' in Figure 3).

The development of global total applied nitrogen over time reflects both changes in the area that nitrogen is applied to as well as changes in the application rates. Figure 4 illustrates this for mineral fertilizer application for three different CFTs. The chosen CFTs reflect historical trends for different crop types from the LUH2 source dataset: 'Temperature cereals' are an example of C3 annual crops in LUH2, maize is a C4 annual crop, while soybean is an N-fixing crop.

Growing areas for rainfed temperate cereals peak during the early 1980s and then decline until the end of the time series. Fertilizer application rates peak around 1985, then decline until the early 2000s and rise again towards the end of the time series, although they do not return to the highs of the 1980s.

Growing areas for rainfed maize rise continuously between 1950 and the end of the time series except for a two small slumps during the early 1980s and early 2000s. Expansion of maize growing areas speeds up towards the end of the time series. Median fertilizer application rates (depicted as circles in Figure 4) peak at 70 kg/ha during the early 1980s, then decline for about 15 years and settle around 61–66 kg/ha after 2000. The 75 % and 90 % quantiles of fertilizer application rates show a more or less continuous increase between 1950 and the end of the time series (triangles and + signs in Figure 4). This suggests that fertilizer use intensified in high-input maize systems over the whole period, but stagnated in lower input systems after the 1980s. Fertilizer application rates for both temperate cereals and maize are very similar between the 5' and 30' dataset.

Even though total growing areas for rainfed soybean are significantly lower than those for maize and temperate cereals, they expand at a much faster rate than the other two crops starting from the early 1960s. Maximum fertilizer application rates for soybean are much higher in the 5' dataset than the 30' dataset until the early 1990s but these high rates are restricted to very small areas and average out in the spatial aggregation to 30'. Median fertilizer application rates for soybean show only small differences between the 5' and 30' dataset: they peak around 1995 at 23 kg/ha in both datasets and then decline to 15 kg/ha and 14 kg/ha, respectively, in the last 5-year time slice. 75 % quantiles of fertilizer application rates rise until the 1990s, then stagnate around 30–32 kg/ha at both spatial resolutions. 90 % quantiles of application rates rise until the early 2000s, then decline again. Overall, fertilizer application rates for soybean are much lower than the rates for maize and temperate cereals. This makes sense because soybean is capable of nitrogen fixation from the atmosphere and therefore requires less additional nitrogen fertilizer.







**Figure 4.** Changes in mineral fertilizer application areas and application rates over time. Each coloured line represents the cumulative growing area of the crop in relation to the respective fertilizer application rate for a 5-year time slice. Symbols illustrate the area-weighted 50 % (median), 75 % and 90 % quantiles of application rates. Top row: rainfed temperate cereals; middle row: rainfed maize; bottom row: rainfed soybean.



## 3.5 Technical notes

We note that some of the data processing in this toolbox is quite resource-intensive. Memory requirements for some of the scripts exceed 32 GB of RAM, especially in the case of the test application at 5' spatial resolution. Scripts that allow parallel computation on multiple CPUs may exceed 32 GB of RAM per CPU. Some R packages such as the 'raster' package can also create substantial amounts of temporary files so care needs to be taken when running on systems with limited disk space assigned to temporary file systems.

We discovered during tests that the result of the shape intersection between GADM polygons and grid cells shows some dependency on the version of the 'sf' package used. The results presented in section 3.1 are based on 'sf' version 0.8-1. Testing the code with a newer version of 'sf' (1.0-3), the shape intersection at 30' spatial resolution returned 12 additional grid cells containing land and assigned two grid cells to a different country. This is because, starting with version 1.0, the 'sf' package uses spherical geometry for operations involving geographic coordinates, based on functionality provided by the 's2' package (Dunnington et al., 2021). Earlier versions of 'sf' use planar geometry for geographic coordinates. There are options in the 'sf' package to switch off 's2' functionality but results are still not completely identical to earlier versions of the package. For the lake and river input, the effect is negligible as global areas covered by lakes and rivers change only by about 1–2 $\mathrm{km}^2$ depending on whether spherical geometry is used or not. This is about two orders of magnitude smaller than the difference between aggregating gridded GLWD level 3 data and aggregating polygon-based GLWD level 1 and 2 data.

We also discovered that soil texture classes and soil pH values derived in section 3.2 show some dependency on package versions. This seems to be related to different versions of the 'raster' package assigning slightly different coordinates to cells in the source raster dataset. As a result, a different dominant soil texture class may be selected in cells where two texture classes have virtually identical area shares. In our test, this affected one grid cell in the 30' dataset and 419 grid cells in the 5' dataset.

We do not recommend specific software versions to be used with the toolbox but caution that created input datasets may not only depend on changes in the source datasets but also on software versions.

## 4 Conclusions

Non-climatic input data for TEMs are often based on various data sources that partially contradict each other and the processing of these data is not trivial and often not well documented. We have developed a toolbox to create basic inputs for TEMs such as LPJmL in a transparent and flexible manner. The toolbox documents the various source datasets and the data processing to combine datasets of multiple types and origins (e.g. gridded, polygon, country statistics) into input datasets for the model. Often, choices need to be made on how to resolve conflicts between datasets and the best solution may depend on the intended usage. Therefore options to modify these choices have been implemented so that the toolbox can be used to generate tailor-made input data for different objectives. Similarly, we present code of the *LandInG* toolbox, so that data can be newly combined if new data sets become available or time series need to be extended. The utility of the toolbox was demonstrated by using it to create two parallel sets of model inputs at spatial resolutions of 5' and 30'. Beyond a simple comparison of the two resolutions,



we have also highlighted the effect of some of the user-settable options and of some interchangeable source datasets on the
1030    resultant model inputs even at the same spatial resolution.

*Code and data availability.*    The code of the toolbox is distributed through https://github.com/PIK-LPJmL/LandInG. The exact code version
used for the sample application is archived at Zenodo under https://doi.org/10.5281/zenodo.7371650 (Ostberg, 2022). The toolbox uses
publicly available datasets for which references are provided in the reference list

*Author contributions.*    S.O. developed and implemented the toolbox, carried out the test application, analysed toolbox outputs and wrote the
1035    paper. C.M. and J.H. contributed data processing components. All co-authors provided comments and guidance on writing the paper.

*Competing interests.*    One of the co-authors is a member of the editorial board of Geoscientific Model Development. The peer-review process
was guided by an independent editor, and the authors have also no other competing interests to declare.

*Acknowledgements.*    This work was supported by the Defense Advanced Research Project Agency under the World Modelers programme
(grant W911NF1910013).





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
