# Peer review of "LandInG 1.0: A toolbox to derive input datasets for terrestrial ecosystem modelling at variable resolutions from heterogeneous sources"

_Geoscientific Model Development, 2022_

## Author Comment (AC1)

*We thank all three referees for their comments, which will help to improve the manuscript. In the following, we reply to each comment individually. All our responses are in italic font.*

**Review by Jinfeng Chang:**
This is a comprehensive manuscript that described a toolbox for generating commonly used input datasets for terrestrial ecosystem modelling at two spatial resolution 5' and 30'. The generated datasets include static inputs like land-sea mask, country and region mask, soil texture and pH, river routing, grid locations of lakes, rivers, dams and reservoirs, and dynamic inputs, a harmonized gridded annual land use and land management (irrigation and fertilization) for the historical period 1500-2017.
The application of this toolbox for generating input datasets for LPJmL was presented as an example. The manuscript is well structed and very well written. I would think it is a valuable effort to facilitate the input generation. I only have a few suggestions as follows.

*Response. We thank Jinfeng Chang for his appreciation of our manuscript.*

- Given the fact that 1) most of the source datasets existed or used in this toolbox has the highest resolution of 5 arc minutes, 2) the spatial resolution of the TEM simulation usually (if not all) depend on the coarse resolution of all input datasets, and 3) in many cases of this toolbox, the aggregation can only be done with an integer multiple of the source resolution, it could be better to give the possible resolutions for each of the input datasets.

*Response: We want to stress that the source datasets used in the example application are only examples. Users can replace any of the source datasets with other datasets that provide comparable information. For example, users can replace the gridded cropland and grazing land from HYDE with a remote-sensing-based land cover dataset at a higher spatial resolution – although this may have a reduced temporal coverage. The land-sea mask and grid (section 2.1), and the country and region mask (section 2.2) can be generated at any grid resolution. By default, LandInG will aggregate soil data (section 2.3) to any integer multiple of the source resolution (30 arc-seconds for the default source dataset), but test users of LandInG have also used it to aggregate to a non-integer multiple resolution. River routing-related model inputs (section 2.4.1) can be generated at any resolution for which the user provides a drainage direction map, so for this input dataset the possible resolutions depend entirely on the resolution of the source dataset. The lake & river input (section 2.4.2) can be generated at any gridded resolution if it is based on polygon source data. The dams and reservoir input (section 2.4.3) can be generated for any gridded resolution for which a drainage direction map is available. The elevation input also required by the reservoir module in LPJmL depends on the resolution of the source dataset. The default source dataset has a spatial resolution of 1 arc-minute, but other digital elevation models at higher spatial resolutions exist. Using the default source datasets and the default assumptions, land use and land management datasets (section 2.5) could be generated at any integer multiple of 5 arc-minute.*

- For all the datasets, it is essential to provide not only the reference, but also the link to the source datasets, the access date (as datasets can be updated), the original data format, and the data content (e.g., exact variable name used by the toolbox). Otherwise, it makes the toolbox much more difficult and less useful for users.

*Response: We note that the documentation (read me) files included in the LandInG code release provide more detailed guidance on how to download the proposed default source datasets and on the formats. As for the exact versions and access dates, one of the goals behind the development of the LandInG toolbox is to enable users to replace source datasets – for example with updated versions – and not be stuck with the versions that were used during the toolbox development. That said, we will add more information about the exact variables used where it is currently missing. For*

*example, we noticed that subsection 2.5.6 on nitrogen fertilizer inputs is a little fuzzy about the variables used from the source datasets.*

- It is understandable that the authors did not provide results on the created gridded maps as it might contain source datasets that require a license to publish. But for those datasets that were publicly available and has been licensed to distribute, it would be better to provide the resulted maps in addition to the code. For the resulted gridded land use and land management dataset, in particular, the strategy described in this manuscript is in some sense novel (or at least comprehensively described for the first time). Though the authors claimed that this manuscript is solely a description of the toolbox, putting the gridded land use and land management dataset into a public repository could be useful for the community.

*Response: The example application of the LandInG toolbox encompasses about 900 GB of data (source data, intermediate and final results). We have considered releasing some of the resulting datasets such as the land use and land management datasets into a public repository but have not found a suitable option so far.*

**Review by Anonymous Referee #2:**
Review of "LandInG 1.0: A toolbox to derive input datasets for terrestrial ecosystem modelling at variable resolutions from heterogeneous sources" by Sebastian Ostberg, Christoph Müller, Jens Heinke, and Sibyll Schaphoff.

In their manuscript, Ostberg et al. describe the recently published version 1.0 of LandInG, focusing on the generation of detailed input data sets for TEMs and describing the algorithms used to derive inputs for LPJmL.

Overall this is a superb manuscript nearly ready for publication. Having used the LPJmL model more than a decade ago, I am thrilled by the improvement in the quality of documenting the input data sets, potentially extendable to other TEMs as well.

The manuscript could well be published as it is, though I have a few suggestions for minor improvements.

*Response: We thank the reviewer for their appreciation of our manuscript.*

One thing I missed when reviewing the manuscript is a table summarizing the input data sets considered, preferably early in section 2. This would allow the reader to gain a quick overview.

*Response: If the reviewer is referring to an overview of the model input data sets that are covered by LandInG, this is already given in Table 1 of the manuscript. If the reviewer is referring to an overview of the source datasets used to generate the model inputs, there is no summary overview for all source datasets, but these are mentioned in each respective section. The top row in Figure 1 summarizes the source datasets used for the land use and land management inputs. We will add a brief overview as an additional column to Table 1.*

I also found some of the highly detailed sections of the manuscript, for example section 2.5.3, somewhat more difficult to follow than other sections of the manuscript. This was largely due to the necessary level of detail, and improvement would be difficult – the authors need to be aware of it, though. One possible improvement might be to enable the reader to still understand the bulk of the document without needing to go into the sections dealing with the details, by one the one hand indicating which sections are safe to skip, and on the other hand by ensuring that no important

information is lost to the reader skipping those sections. However, this may be more effort than worthwhile, so I leave it to the authors to decide.

*Response: Referee #3 suggested to state a rationale for why the different source datasets are combined before going into the details of how the datasets are combined. Adding this may enable the reader to get the general idea without having to read all the details.*

I thank the authors for the care taken in copy-editing, as there were nearly zero spelling or grammar errors to be found in the manuscript, something a reviewer unfortunately cannot take for granted.

*Response: Thank you very much.*

Finally, the reader subconsciously expects to see some maps. Is there really nothing worthwhile showing in map form? Maybe the authors find one or two examples from section 3 that illustrate current capabilities or improvements in comparison to previous approaches? Being unfamiliar with the exact output, I cannot make specific suggestions, however I did wonder what difference the choice in land-sea mask makes in comparison to CRU, and in a number of places I would have liked to see changes in comparison to Schaphoff18, but I know most of those maps will be rather boring due to the few changed points and small magnitude of changes, so I again leave it to the authors to decide.

*Response: We thank the reviewer for the suggestion. Indeed, many of the maps may be rather boring, because, for example, the land-sea mask is identical over the vast majority of the land surface and only differs along coastlines or for large inland water bodies such as the Caspian Sea. Other inputs such as the river routing or the neighbour cell network for irrigation cannot be interpreted visually because they contain cell indices. For land use, fertilizer and manure application datasets, the LandInG toolbox creates up to 518 annual patterns for a total of 32 land use classes, so any single example would have to be a rather arbitrary selection. We will add maps comparing the two soil aggregation algorithms, which show visible differences.*

**Review by Anonymous Referee #3:**
General comments
The manuscript by Ostberg et al. presents a tool aiming at harmonizing an ensemble of heterogeneous datasets in order to generate consistent input data for terrestrial ecosystem models. The data considered in the study are very diverse, including information relative to land-sea mask, river routing, dams, land use, nitrogen fertilizers, ...
In addition to the description of the methods developed, the manuscript reports on the application of the tool at two spatial resolutions (5' and 30') and evaluates how the data generated through the harmonization procedure compare to the raw data and/or to other reference datasets. Both sections dedicated to the description of the tool (section 2) and to its evaluation (section 3) are relevant and highly detailed.
All land modelling groups are faced to the problem of harmonizing input raw data. This is done internally into the model or externally but often quickly and in a more or less clean way, and is rarely – if not never – documented. In this respect, the tool developed by Ostberg et al. is very welcome for the land modelling community and the efforts for accurately describing / documenting the tool in the manuscript should be acknowledged.

*Response: We thank the reviewer for their appreciation of our manuscript.*

The manuscript fits well into the scope of GMD journal (although ESSD, another Copernicus journal could have been envisaged to my opinion) and the description of the LandInG merits

certainly to be published. However, some rewriting or reshaping should be envisaged to ease the reading of the manuscript. Here below are some points, which could be improved to my opinion.

*Response: Regarding the journal choice, we specifically decided for GMD instead of, e.g. ESSD, because our main goal is to publish the toolbox itself rather than the datasets created using it so that other users can re-use the code, create updated versions or tailor some of the assumptions made to their specific applications or their models.*

Specific comments
The LandInG has been used for LPJmL but is generic enough to be used for other TEMs, with possibly some changes. The reading of the manuscript gives the impression that the authors don't want to provide too many information specific to LPJmL about input requirement because of the genericity of the tool. As a consequence, it is not clear what are the input data needed by LPJmL and that need to be generated by the tool. To my opinion, the manuscript will gain assuming the toolbox has been used so far for LPJmL input data and to base the description of the methods on this model solely. It does not remove anything to the genericity of the tool. The LPJmL input data description could be done more clear at the beginning of each subsection (2.2, 2.3, 2.4.1, 2.4.2, ...). This is a detail but if focusing only on LPJmL context, formulations such as "TEMs such as LPJmL" (line 93, 259, 271, 274) could be removed of the section 2.

*Response: The input datasets described in the manuscript indeed include all inputs other than climate inputs needed to run LPJmL in standard applications. However, not all of the datasets are required to run LPJmL depending on the model configuration and specific inputs needed for specific applications are not covered by LandInG (e.g. tillage type data). The most basic setup for LPJmL only requires the grid and soil inputs from this toolbox (section 2.1 and 2.3). We expect that other TEMs can also be run in multiple configurations with different needs in terms of inputs. We will revise the text towards better readability and avoid repetitive use of "TEMs such as LPJmL" or similar.*

A specific section relative to the Application of the LandInG toolbox for other TEMs could be envisaged at the end of the manuscript or at the end of section 2. It would lighten the description of the method while gathering all the information about the genericity and possible further development envisaged to gain in genericity, in a specific section. The attempt is not to list all input data needed by any TEM, but to identify some of them for which an update of LandInG would be needed. For instance, one about the description of natural vegetation, which is computed internally in LPJmL but often prescribed in many models (this point is mentioned lines 275-280). Another feature may concern the soil information, since some models (see for instance Chaney et al., 2018 https://doi.org/10.5194/hess-22-3311-2018) start setting soil properties at the tile level and not only at the grid cell level.

*Response: As stated above, LandInG is not tailor made for LPJmL even though of course LPJmL applications are a central motivation for the LandInG development. Instead of speculating what inputs may be missing for other models or applications of any TEM, we will check the introduction section for better clarity on the genericity and special application to LPJmL.*

Some information on the general rationale behind all the data processing would be useful prior describing the different steps done for generating input data for LPJmL (sections 2.1 to 2.5). For example, in section 2.5 on "land use and land management", subsection 2.5.1 focuses on "country level source data" but the authors do not explain first why such country-scale data are needed. To my opinion, the authors should explain in few sentences in the first paragraph of section 2.5 and before subsection 2.5.1 that some data provide information for hundreds of crop types but only at country scale (FAOSTAT) while other data are gridded but provide only total cropland area for

instance (HYDE). The authors want to take advantage of both. I think explaining this kind rationale first, prior going in all the details of the data processing, would be useful to the reader and not only for the data about "land use and land management" but for any kind of data (from section 2.1 to 2.5).

*Response: Thank you for the suggestion. We agree that explaining the rationale for combining the various source datasets into the land use and land management inputs in section 2.5 is a useful addition to the manuscript. As for sections 2.1. to 2.4, these input datasets are usually derived from a single source dataset, making the link from source to input dataset clear. Also, each subsection already includes an explanation of the purpose of these input datasets for the model simulations.*

Although all the content of section 3 is of value, this section is quite long and is not always ease to read. I would suggest to shorten it and to limit it to the key results about the application of the tool at 5' and 30' resolution. If needed, part of the materials and of the results could be moved in Appendices. Similarly the section on 'Technical notes' could be moved to my opinion in a Appendix. If not using Appendices, I would encourage the authors to add an additional level to the subsection (3.X.X.X) in order to better structure this section and facilitate its reading.

*Response: The structure of section 3 mostly follows the structure of section 2 and we see little additional guidance in a fourth level of subsections. We will add text at the beginning of section 3 emphasizing that this section does not introduce new functionality but only illustrates implications of choices made in the data processing, which can certainly be skipped by readers not interested in this aspect. We would like to keep the section (including subsection 3.5) in the main text, however, as we think that the differences described illustrate very well how important specific choices (such as the spatial resolution) can be for the final product.*

In the Introduction section (line 22), the authors should add information on the objectives of the toolbox and of the manuscript, prior to detail what the sections contain. The name of the toolbox, in short (LandInG) or long name (Land Input Generator) is not even mentioned in the Introduction. Regarding the objectives, the abstract is more informative than the introduction itself.

*Response: Thank you for the suggestion. We agree that the abstract currently contains some information that should also be included in the introduction. We will expand the introduction.*

Minor comments / Technical corrections
To my opinion, the naming of section 2 and subsections should be related to the data processes and harmonization procedures developed for generating the input data of LPJmL and not related to the raw data used, as it is in the current manuscript.

*Response: Section 2 is structured based on different types of model inputs that can be created using the LandInG toolbox. Since not all users may need all types of inputs for their respective models or applications, we think that this structure makes more sense than ordering the section by different types of source datasets (e.g. polygon versus gridded versus census datasets) or different processes of data harmonization.*

Line 44: "do not seem to contain any land". Could you be more affirmative?

*Response: We will change the sentence to "do not contain any land from a visual inspection of satellite data with Google Maps"*

Line 178: I would suggest replacing "land" by "other land categories"

*Response: Thank you for the suggestion. We will change the sentence accordingly.*

Line 197: could you specify what the elevation above sea level is used for in LPJmL?

*Response: Elevation above sea level is used as one criterion to determine which cells may withdraw irrigation water from reservoirs in the model. We will add this information to the manuscript.*

Line 198-199 : could you detail what are the data from GranD you used in the toolbox before mentioning (lines 200-201) that data without storage capacity or reservoir area are removed.

*Response: The information about which data from GranD are used is given in lines 226-234. It is probably a good idea to mention this earlier in the subsection. We will add a sentence before detailing how the data are processed.*

Figure 1, page 11: Could you specify on top of the figure, what are the data you use from the different datasets (GAEZ, HYDE, MON, ...), below the green and blue boxes.

*Response: The figure is already very dense, and depending on the source dataset, the list of used variables can be quite long (e.g. 6 agro-climatic variables from GAEZ, more than 10 variables from LUH2). Adding all of these to the figure would make it unreadable. The text provides more details on the used variables. However, we noticed that section 2.5.6 is a little fuzzy on the used variables. We will add the missing information there.*

Lines 322-323: "Since the cropland assumptions underlying MON differ from the HYDE cropland used here". Could you say more on this?

*Response: Cropland patterns for the year 2000 used to create the MON dataset differ from cropland patterns from HYDE for the same year that we use in the LandInG toolbox. We will rephrase the sentence in line 322-324 to make that clearer.*

Line 323: "total" instead of "global" ?

*Response: RAM is a global gridded cropland dataset for the year 2000 that was used to create MON. As mentioned in our response to the previous comment, we will rephrase the sentence.*

Line 657: move the reference to Table 4 after "135 106 km2"
Line 657: could you give a reference for the estimation of "3% of ocean" in the land area estimate?

*Response: There is no literature reference for the "3% of ocean". This is derived as the difference between total land area and total grid area in Table 4, which is why the reference to Table 4 is at the end of the sentence.*

Legend of Figure 2: "Dashed lines indicate the part of total and irrigated harvested areas that are based on gapfilling or extrapolation of country-level source data." This is not so ease to visualize and to understand why some lines are vertical, and another covers all the light blue area. The dashed lines on the dark blue are almost not visible.

*Response: We acknowledge that this is not easy to understand. Because of the sparsity of source data on irrigated harvested areas, 100% of irrigated harvested areas can be based on extrapolation or gapfilling in some years, whereas that number is much lower in years that exist in the source data. We had hoped that the description in the figure caption was clear enough but it seems that is*

*not. Since this information is contained in the main text, we think it is probably best to remove the dashed lines from the figure completely.*

Line 863-864: "The toolbox designates any cropland area that exceeds the sum of crop-specific growing areas in a grid cell as fallow land." I think this information should be moved in section 2 where you describe the toolbox itself.

*Response: We explain in section 2.5.5 (lines 581– 583) how fallow land is determined. The sentence in line 863–864 is only to remind the reader. This is not new information that is introduced at that point. We will add a cross-reference to section 2.5.5. so that it is clear that this is described in more detail there.*

Figure 3: Y-axis unit, replace "TgN" by "TgN yr-1"

*Response: We will change the Y-axis label.*

Line 954: "Teragrams N per year (TgN yr-1)" instead of "Teragrams (Tg)"
Line 955: "TgN yr-1" instead of "Tg" (the same lines 958 and 962)

*Response: We will change the units in the text in lines 954, 955, 958, and 962.*

Figure 4: add "N" in the x-axis unit "(kgN/ha)"

*Response: We will change the X-axis label.*

---

## Author Response (AR1)

Dear Yilong Wang,

Please find attached the revised version of our manuscript "LandInG 1.0: A toolbox to derive input datasets for terrestrial ecosystem modelling at variable resolutions from heterogeneous sources".

We have responded to all comments made by the three referees and updated the manuscript in a number of places. Below, you will find the referee comments with our replies in italic and a list of changes made to the manuscript. We also attach a marked-up version of the manuscript showing all changes. All line numbers mentioned in our response refer to the marked-up version.

We believe that the changes have improved the manuscript and hope that it is now in a state suitable for publication.

Best regards,
Sebastian Ostberg

**Review by Jinfeng Chang:**
This is a comprehensive manuscript that described a toolbox for generating commonly used input datasets for terrestrial ecosystem modelling at two spatial resolution 5' and 30'. The generated datasets include static inputs like land-sea mask, country and region mask, soil texture and pH, river routing, grid locations of lakes, rivers, dams and reservoirs, and dynamic inputs, a harmonized gridded annual land use and land management (irrigation and fertilization) for the historical period 1500-2017.
The application of this toolbox for generating input datasets for LPJmL was presented as an example. The manuscript is well structed and very well written. I would think it is a valuable effort to facilitate the input generation. I only have a few suggestions as follows.

*Response. We thank Jinfeng Chang for his appreciation of our manuscript.*

- Given the fact that 1) most of the source datasets existed or used in this toolbox has the highest resolution of 5 arc minutes, 2) the spatial resolution of the TEM simulation usually (if not all) depend on the coarse resolution of all input datasets, and 3) in many cases of this toolbox, the aggregation can only be done with an integer multiple of the source resolution, it could be better to give the possible resolutions for each of the input datasets.

*Response: We want to stress that the source datasets used in the example application are only examples. Users can replace any of the source datasets with other datasets that provide comparable information. For example, users can replace the gridded cropland and grazing land from HYDE with a remote-sensing-based land cover dataset at a higher spatial resolution – although this may have a reduced temporal coverage. Regarding the possible resolutions of the input datasets: The land-sea mask and grid (section 2.1), and the country and region mask (section 2.2) are based on the same source dataset and can be generated at any grid resolution, as mentioned in lines 61–62. By default, LandInG will aggregate soil data (section 2.3) to any integer multiple of the source resolution (30 arc-seconds for the default source dataset), but test users of LandInG have also used it to aggregate to a non-integer multiple resolution. River routing-related model inputs (section 2.4.1) can be generated at any resolution for which the user provides a drainage direction map, so for this input dataset the possible resolutions depend entirely on the resolution of the source dataset, as mentioned in lines 161–162. Possible resolutions for the lake & river input (section 2.4.2) are mentioned in lines 202–206. The dams and reservoir input (section 2.4.3) can be generated for any gridded resolution for which a drainage direction map is available. The elevation input also required by the reservoir module in LPJmL depends on the resolution of the source*

*dataset. The default source dataset has a spatial resolution of 1 arc-minute, but other digital elevation models at higher spatial resolutions exist. Using the default source datasets and the default assumptions, land use and land management datasets (section 2.5) could be generated at any integer multiple of 5 arc-minute, as mentioned in lines 312–314.*

*We added a new sentence to section 2.4.3 (lines 215–216) clarifying that the reservoir input requires the river routing input from section 2.4.1.*

- For all the datasets, it is essential to provide not only the reference, but also the link to the source datasets, the access date (as datasets can be updated), the original data format, and the data content (e.g., exact variable name used by the toolbox). Otherwise, it makes the toolbox much more difficult and less useful for users.

*Response: We note that the documentation (read me) files included in the LandInG code release provide more detailed guidance on how to download the proposed default source datasets and on the formats. As for the exact versions and access dates, one of the goals behind the development of the LandInG toolbox is to enable users to replace source datasets – for example with updated versions – and not be stuck with the versions that were used during the toolbox development.*

*That said, we have added the exact variables and/or file formats used in the respective sections (lines 117, 118, 219–222, 227, 228, 286,  341–342, 636–639, 652, 655–656, 695–696, 711, 754–755, 806, 810, 818) of the revised manuscript.*
*We also added missing download links and access dates to the reference list (lines 1114, 1119, 1158, 1167–1168 1178, 1180, 1185, 1197, 1999, 1222–1223, 1228, 1252–1253, 1278–1280).*

- It is understandable that the authors did not provide results on the created gridded maps as it might contain source datasets that require a license to publish. But for those datasets that were publicly available and has been licensed to distribute, it would be better to provide the resulted maps in addition to the code. For the resulted gridded land use and land management dataset, in particular, the strategy described in this manuscript is in some sense novel (or at least comprehensively described for the first time). Though the authors claimed that this manuscript is solely a description of the toolbox, putting the gridded land use and land management dataset into a public repository could be useful for the community.

*Response: The example application of the LandInG toolbox encompasses about 900 GB of data (source data, intermediate and final results). We have considered releasing some of the resulting datasets such as the land use and land management datasets into a public repository that should ideally also offer some data discovery and visualization functionality but have not found a suitable option so far. Also, some of the source datasets used in the sample application have been updated in the meantime, so users may want to re-run the toolbox with the updated source datasets.*

**Review by Anonymous Referee #2:**
Review of "LandInG 1.0: A toolbox to derive input datasets for terrestrial ecosystem modelling at variable resolutions from heterogeneous sources" by Sebastian Ostberg, Christoph Müller, Jens Heinke, and Sibyll Schaphoff.

In their manuscript, Ostberg et al. describe the recently published version 1.0 of LandInG, focusing on the generation of detailed input data sets for TEMs and describing the algorithms used to derive inputs for LPJmL.

Overall this is a superb manuscript nearly ready for publication. Having used the LPJmL model more than a decade ago, I am thrilled by the improvement in the quality of documenting the input data sets, potentially extendable to other TEMs as well.

The manuscript could well be published as it is, though I have a few suggestions for minor improvements.

*Response: We thank the reviewer for their appreciation of our manuscript.*

One thing I missed when reviewing the manuscript is a table summarizing the input data sets considered, preferably early in section 2. This would allow the reader to gain a quick overview.

*Response: If the reviewer is referring to an overview of the model input datasets that are covered by LandInG, this is already given in Table 1 of the manuscript. If the reviewer is referring to an overview of the source datasets used to generate the model inputs, there is no summary overview for all source datasets, but these are mentioned in each respective section. The top row in Figure 1 summarizes the source datasets used for the land use and land management inputs.*

*We have added a brief overview of the source datasets as a third column to Table 1 but still refer the reader to the respective sections for the full list of source datasets.*

I also found some of the highly detailed sections of the manuscript, for example section 2.5.3, somewhat more difficult to follow than other sections of the manuscript. This was largely due to the necessary level of detail, and improvement would be difficult – the authors need to be aware of it, though. One possible improvement might be to enable the reader to still understand the bulk of the document without needing to go into the sections dealing with the details, by one the one hand indicating which sections are safe to skip, and on the other hand by ensuring that no important information is lost to the reader skipping those sections. However, this may be more effort than worthwhile, so I leave it to the authors to decide.

*Response: Referee #3 suggested to state a rationale for why the different source datasets are combined before going into the details of how the datasets are combined. We believe that this may also enable the reader to get the general idea without having to read all the details.*

*We have added statements explaining the rationale behind certain processing steps before going into the details in lines 370–372, 503–504, 538–539).*

I thank the authors for the care taken in copy-editing, as there were nearly zero spelling or grammar errors to be found in the manuscript, something a reviewer unfortunately cannot take for granted.

*Response: Thank you very much.*

Finally, the reader subconsciously expects to see some maps. Is there really nothing worthwhile showing in map form? Maybe the authors find one or two examples from section 3 that illustrate current capabilities or improvements in comparison to previous approaches? Being unfamiliar with the exact output, I cannot make specific suggestions, however I did wonder what difference the choice in land-sea mask makes in comparison to CRU, and in a number of places I would have liked to see changes in comparison to Schaphoff18, but I know most of those maps will be rather

boring due to the few changed points and small magnitude of changes, so I again leave it to the authors to decide.

*Response: We thank the reviewer for the suggestion. Indeed, many of the maps may be rather boring, because, for example, the land-sea mask is identical over the vast majority of the land surface and only differs along coastlines or for large inland water bodies such as the Caspian Sea. Other inputs such as the river routing or the neighbour cell network for irrigation cannot be interpreted visually because they contain cell indices. For land use, fertilizer and manure application datasets, the LandInG toolbox creates up to 518 annual patterns for a total of 32 land use classes, so any single example would have to be a rather arbitrary selection.*

*We have added a new Figure 2 to the revised manuscript with two maps comparing the two soil aggregation algorithms, which show visible differences. The figure is referred to in line 749 of the text.*

**Review by Anonymous Referee #3:**
General comments
The manuscript by Ostberg et al. presents a tool aiming at harmonizing an ensemble of heterogeneous datasets in order to generate consistent input data for terrestrial ecosystem models. The data considered in the study are very diverse, including information relative to land-sea mask, river routing, dams, land use, nitrogen fertilizers, ...
In addition to the description of the methods developed, the manuscript reports on the application of the tool at two spatial resolutions (5' and 30') and evaluates how the data generated through the harmonization procedure compare to the raw data and/or to other reference datasets. Both sections dedicated to the description of the tool (section 2) and to its evaluation (section 3) are relevant and highly detailed.
All land modelling groups are faced to the problem of harmonizing input raw data. This is done internally into the model or externally but often quickly and in a more or less clean way, and is rarely – if not never – documented. In this respect, the tool developed by Ostberg et al. is very welcome for the land modelling community and the efforts for accurately describing / documenting the tool in the manuscript should be acknowledged.

*Response: We thank the reviewer for their appreciation of our manuscript.*

The manuscript fits well into the scope of GMD journal (although ESSD, another Copernicus journal could have been envisaged to my opinion) and the description of the LandInG merits certainly to be published. However, some rewriting or reshaping should be envisaged to ease the reading of the manuscript. Here below are some points, which could be improved to my opinion.

*Response: Regarding the journal choice, we specifically decided for GMD instead of, e.g. ESSD, because our main goal is to publish the toolbox itself rather than the datasets created using it so that other users can re-use the code, create updated versions or tailor some of the assumptions made to their specific applications or their models.*

Specific comments
The LandInG has been used for LPJmL but is generic enough to be used for other TEMs, with possibly some changes. The reading of the manuscript gives the impression that the authors don't want to provide too many information specific to LPJmL about input requirement because of the genericity of the tool. As a consequence, it is not clear what are the input data needed by LPJmL

and that need to be generated by the tool. To my opinion, the manuscript will gain assuming the toolbox has been used so far for LPJmL input data and to base the description of the methods on this model solely. It does not remove anything to the genericity of the tool. The LPJmL input data description could be done more clear at the beginning of each subsection (2.2, 2.3, 2.4.1, 2.4.2, ...). This is a detail but if focusing only on LPJmL context, formulations such as "TEMs such as LPJmL" (line 93, 259, 271, 274) could be removed of the section 2.

*Response: The input datasets described in the manuscript indeed include all inputs other than climate inputs needed to run LPJmL in standard applications. However, not all of the datasets are required to run LPJmL depending on the model configuration and specific inputs needed for specific applications are not covered by LandInG (e.g. tillage type data). The most basic setup for LPJmL only requires the grid and soil inputs from this toolbox (section 2.1 and 2.3). We expect that other TEMs can also be run in multiple configurations with different needs in terms of inputs.*

*We have added text explaining which input datasets are required by LPJmL under which conditions in the respective subsections (lines 42–43, 86, 155, 198, 215–216).*
*We have removed repetitive use of "TEMs such as LPJmL" (lines 109, 292, 295, 1070).*

A specific section relative to the Application of the LandInG toolbox for other TEMs could be envisaged at the end of the manuscript or at the end of section 2. It would lighten the description of the method while gathering all the information about the genericity and possible further development envisaged to gain in genericity, in a specific section. The attempt is not to list all input data needed by any TEM, but to identify some of them for which an update of LandInG would be needed. For instance, one about the description of natural vegetation, which is computed internally in LPJmL but often prescribed in many models (this point is mentioned lines 275-280). Another feature may concern the soil information, since some models (see for instance Chaney et al., 2018 https://doi.org/10.5194/hess-22-3311-2018) start setting soil properties at the tile level and not only at the grid cell level.

*Response: As stated above, LandInG is not tailor made for LPJmL even though of course LPJmL applications are a central motivation for the LandInG development. We do not want to speculate in detail about what inputs may be missing for other models or applications of any TEM.*

*We have still added wording in a few places to point out details of the toolbox that are specific to LPJmL and how this may be different for other TEMs (lines 41–47, 112, 603).*

Some information on the general rationale behind all the data processing would be useful prior describing the different steps done for generating input data for LPJmL (sections 2.1 to 2.5). For example, in section 2.5 on "land use and land management", subsection 2.5.1 focuses on "country level source data" but the authors do not explain first why such country-scale data are needed. To my opinion, the authors should explain in few sentences in the first paragraph of section 2.5 and before subsection 2.5.1 that some data provide information for hundreds of crop types but only at country scale (FAOSTAT) while other data are gridded but provide only total cropland area for instance (HYDE). The authors want to take advantage of both. I think explaining this kind rationale first, prior going in all the details of the data processing, would be useful to the reader and not only for the data about "land use and land management" but for any kind of data (from section 2.1 to 2.5).

*Response: Thank you for the suggestion. We agree that explaining the rationale behind combining the various source datasets into the land use and land management inputs in section 2.5 is a useful*

*addition to the manuscript. As for sections 2.1. to 2.4, these input datasets are usually derived from a single source dataset, making the link from source to input dataset clear. Also, each subsection already includes an explanation of the purpose of these input datasets for the model simulations.*

*We have added wording explaining the rationale behind combining the different source datasets (lines 305–310, 321–322, 339–341, 370–372, 503–504).*

Although all the content of section 3 is of value, this section is quite long and is not always ease to read. I would suggest to shorten it and to limit it to the key results about the application of the tool at 5' and 30' resolution. If needed, part of the materials and of the results could be moved in Appendices. Similarly the section on 'Technical notes' could be moved to my opinion in a Appendix. If not using Appendices, I would encourage the authors to add an additional level to the subsection (3.X.X.X) in order to better structure this section and facilitate its reading.

*Response: The structure of section 3 mostly follows the structure of section 2 and we see little additional guidance in a fourth level of subsections. This section does not introduce new functionality but only illustrates implications of choices made in the data processing, which can certainly be skipped by readers not interested in this aspect. We would like to keep the section (including subsection 3.5) in the main text, however, as we think that the differences described illustrate very well how important specific choices (such as the spatial resolution) can be for the final product.*

*We have added text at the beginning of section 3 emphasizing that this section does not introduce new functionality (lines 689–690).*

In the Introduction section (line 22), the authors should add information on the objectives of the toolbox and of the manuscript, prior to detail what the sections contain. The name of the toolbox, in short (LandInG) or long name (Land Input Generator) is not even mentioned in the Introduction. Regarding the objectives, the abstract is more informative than the introduction itself.

*Response: Thank you for the suggestion.*

*We have expanded the introduction to include the name of the toolbox and to clarify the objectives of the toolbox (lines 22–29).*

Minor comments / Technical corrections
To my opinion, the naming of section 2 and subsections should be related to the data processes and harmonization procedures developed for generating the input data of LPJmL and not related to the raw data used, as it is in the current manuscript.

*Response: Section 2 is structured based on different types of model inputs that can be created using the LandInG toolbox. Since not all users may need all types of inputs for their respective models or applications, we think that this structure makes more sense than ordering the section by different types of source datasets (e.g. polygon versus gridded versus census datasets) or different processes of data harmonization.*

Line 44: "do not seem to contain any land". Could you be more affirmative?

*Response: We have changed the sentence to "do not contain any land from a visual inspection of satellite data with Google Maps" (line 59).*

Line 178: I would suggest replacing "land" by "other land categories"

*Response: Thank you for the suggestion. We have changed the sentence accordingly (line 196).*

Line 197: could you specify what the elevation above sea level is used for in LPJmL?

*Response: Elevation above sea level is used as one criterion to determine which cells may withdraw irrigation water from reservoirs in the model.*

*We have added this information to the manuscript (275–276).*

Line 198-199 : could you detail what are the data from GranD you used in the toolbox before mentioning (lines 200-201) that data without storage capacity or reservoir area are removed.

*Response: We have added a sentence listing the GranD variables used before detailing how the data are processed (lines 218–220).*

Figure 1, page 11: Could you specify on top of the figure, what are the data you use from the different datasets (GAEZ, HYDE, MON, ...), below the green and blue boxes.

*Response: The figure is already very dense, and depending on the source dataset, the list of used variables can be quite long (e.g. 6 agro-climatic variables from GAEZ, more than 10 variables from LUH2). Adding all of these to the figure would make it unreadable. The text provides more details on the used variables.*
*As also requested by the other referee, we have expanded the text in a number of places to add information about the exact variables used (lines 341–342, 636–639, 652, 655–656).*

Lines 322-323: "Since the cropland assumptions underlying MON differ from the HYDE cropland used here". Could you say more on this?

*Response: Cropland patterns for the year 2000 used to create the MON dataset differ from cropland patterns from HYDE for the same year that we use in the LandInG toolbox.*

*We have changed the sentence to clarify "The gridded cropland extent used to develop MON differs from the HYDE cropland used here, so the gridded cropland dataset underlying MON (Ramankutty et al., 2008, referred to as RAM) is used as well to resolve inconsistencies." (lines 353–355).*

Line 323: "total" instead of "global" ?

*Response: RAM is a global gridded cropland dataset for the year 2000 that was used to create MON.*
*As mentioned in our response to the previous comment, we have changed the sentence (lines 353–355).*

Line 657: move the reference to Table 4 after "135 106 km2"

Line 657: could you give a reference for the estimation of "3% of ocean" in the land area estimate?

*Response: There is no literature reference for the "3% of ocean". This is derived as the difference between total land area and total grid area in Table 4, which is why the reference to Table 4 is at the end of the sentence.*

*We have changed "Taking into account the land fraction in each cell ..." into "Taking into account the calculated land fraction in each cell…" to clarify that this a result from the toolbox (line 699).*

Legend of Figure 2: "Dashed lines indicate the part of total and irrigated harvested areas that are based on gapfilling or extrapolation of country-level source data." This is not so ease to visualize and to understand why some lines are vertical, and another covers all the light blue area. The dashed lines on the dark blue are almost not visible.

*Response: We acknowledge that this is not easy to understand. Because of the sparsity of source data on irrigated harvested areas, 100% of irrigated harvested areas can be based on extrapolation or gap-filling in some years, whereas that number is much lower in years that exist in the source data. We had hoped that the description in the figure caption was clear enough but it seems that it was not.*

*We have removed the dashed lines from Figure 3 (what used to be Figure 2 in the first submission) and the figure caption. The values for total and irrigated harvested areas that are based on gap-filling or extrapolation are still mentioned in the text (lines 887, 890) but no longer refer to the figure.*

Line 863-864: "The toolbox designates any cropland area that exceeds the sum of crop-specific growing areas in a grid cell as fallow land." I think this information should be moved in section 2 where you describe the toolbox itself.

*Response: We explain in section 2.5.5 how fallow land is determined. The sentence in line section 3.4.1 is only to remind the reader. This is not new information that is introduced at that point.*

*We have added a cross-reference to section 2.5.5. in line 912 so that it is clear that this is described in more detail earlier.*

Figure 3: Y-axis unit, replace "TgN" by "TgN yr-1"

*Response: We have updated Figure 4 (what used to be Figure 3 in the previous version).*

Line 954: "Teragrams N per year (TgN yr-1)" instead of "Teragrams (Tg)"

Line 955: "TgN yr-1" instead of "Tg" (the same lines 958 and 962)

*Response: We have changed the units in the text (lines 1003–1005, 1008, 1011).*

Figure 4: add "N" in the x-axis unit "(kgN/ha)"

*Response: We have changed the X-axis label in Figure 5 (what used to be Figure 4). We have also changed the units in the text (lines 679, 1007, 1017, 103, 1032, 1040–1042).*

*Besides the changes listed above, which were requested by the referees, we have made some additional changes:*

- *We have added the information that fallow land can be included in the land use input created by the toolbox (line 303).*
- *We have removed duplicate URLs in a few references (lines 1119, 1122, 1163).*
- *We have updated the archived LandInG code at Zenodo to reflect the changes to Figure 3 and Figure 5 and the addition of the new Figure 2. We have also corrected spelling mistakes and updated several download links in the README files included in the LandInG code archive. The manuscript now refers to the updated code archive (lines 1082, 1211).*
- *We have also updated download links and access dates in the reference list included in the Supporting Information (lines 3, 6, 9, 11, 14–15 of the marked-up Supporting Information).*